# Disordered breathing in a mouse model of Dravet syndrome

**Fu-Shan Kuo, Colin M Cleary, Joseph J LoTurco, Xinnian Chen, Daniel K Mulkey***

Department of Physiology and Neurobiology, University of Connecticut, Storrs, United States

**Abstract** Dravet syndrome (DS) is a form of epilepsy with a high incidence of sudden unexpected death in epilepsy (SUDEP). Respiratory failure is a leading cause of SUDEP, and DS patients' frequently exhibit disordered breathing. Despite this, mechanisms underlying respiratory dysfunction in DS are unknown. We found that mice expressing a DS-associated *Scn1a* missense mutation (A1783V) conditionally in inhibitory neurons (*Slc32a1*$^{cre/+}$::*Scn1a*$^{A1783V\ fl/+}$; defined as *Scn1a*$^{\Delta E26}$) exhibit spontaneous seizures, die prematurely and present a respiratory phenotype including hypoventilation, apnea, and a diminished ventilatory response to $CO_2$. At the cellular level in the retrotrapezoid nucleus (RTN), we found inhibitory neurons expressing the *Scn1a* A1783V variant are less excitable, whereas glutamatergic chemosensitive RTN neurons, which are a key source of the $CO_2/H^+$-dependent drive to breathe, are hyper-excitable in slices from *Scn1a*$^{\Delta E26}$ mice. These results show loss of *Scn1a* function can disrupt respiratory control at the cellular and whole animal levels.
DOI: https://doi.org/10.7554/eLife.43387.001

## Introduction

Dravet syndrome (DS) (aka. severe myoclonic epilepsy of infancy) is a severe form of early-onset epilepsy that is resistant to anti-epileptic drugs and has a high incidence of sudden unexpected death in epilepsy (SUDEP) (*Kalume, 2013*; *Kearney, 2013*; *Shmuely et al., 2016*). Mechanisms contributing to SUDEP involve disruption of cardiac and/or respiratory function (*Massey et al., 2014*); the relative contribution of each may vary depending on multiple pathological factors including severity of symptoms and underlying cause of epilepsy. With regard to SUDEP in DS, most work implicates cardiac failure caused by seizure-induced parasympathetic suppression of cardiac activity (*Kearney, 2013*; *Kalume et al., 2013*; *Gataullina and Dulac, 2017*). However, recent evidence suggests respiratory dysfunction precipitates cardiac failure and contributes to mortality in DS. For example, DS patients showed peri-ictal breathing problems including hypoventilation and apnea prior to the manifestation of bradycardia, a slower than normal heart rate (*Kim et al., 2018*). Patients with DS also exhibited a blunted ventilatory response to $CO_2$ (*Kim et al., 2018*). This finding suggests that respiratory dysfunction, possibly at the level of respiratory chemoreceptors (neurons that regulate breathing in response to changes in tissue $CO_2/H^+$), contributes to the pathology of DS. Despite this physiological significance, mechanisms underlying respiratory dysfunction in DS or epilepsy in general are not well understood. Leading hypotheses propose that seizure activity disrupts respiratory control by a feed-forward mechanisms involving spreading depolarization (*Aiba and Noebels, 2015*) or activation of inhibitory subcortical projections to brainstem respiratory centers (*Dlouhy et al., 2015*; *Lacuey et al., 2017*). Consistent with the latter possibility, there is evidence that activity of serotonergic neurons in the dorsal and medullary raphe regions in rats are suppressed during ictal and post-ictal periods (*Zhan et al., 2016*). Serotonin is a potent modulator of breathing and arousal (*Richerson, 2004*; *Buchanan and Richerson, 2010*); therefore, it is possible that loss of this drive during seizures contributes to SUDEP. This possibility is supported by evidence that pharmacological

*For correspondence:
daniel.mulkey@uconn.edu

Competing interests: The authors declare that no competing interests exist.

augmentation of serotonergic signaling can prevent seizure-induced respiratory arrest in a mouse model of epilepsy and may improve seizure control in DS patients (*Tupal and Faingold, 2019*). However, some epilepsy patients show breathing abnormalities under baseline inter-ictal conditions including a reduced ventilatory response to $CO_2$ (i.e., chemoreflex) (*Sainju et al., 2019*), suggesting factors other than seizure activity compromise respiratory control. Based on this, we consider a yet unexplored possibility that epilepsy-associated mutations may directly affect brainstem respiratory centers to compromise breathing under inter-ictal conditions, and thus serve as a common substrate for both seizures and respiratory dysfunction.

Most DS cases (70–95%) are caused by mutations in the *Scn1a* gene (MIM#182389), which encodes the pore-forming subunit of a voltage-gated $Na^+$ channel (Nav1.1) (*Meisler and Kearney, 2005*; *Fujiwara, 2006*; *Catterall et al., 2010*; *Akiyama et al., 2012*). Approximately 700 different *Scn1a* pathological variants have been identified in DS patients, the majority of which are missense or frameshift mutations that result in loss of function (*Parihar and Ganesh, 2013*). Consistent with this, conventional *Scn1a* knockout mouse models (on a mixed C57B/6 background) recapitulate characteristic features of DS, including motor problems, seizures and premature death, in a remarkably titratable manner. For example, homozygous *Scn1a* knockout mice develop ataxia and die at 15 days postnatal, whereas heterozygous *Scn1a* deficient mice show seizure activity and early mortality starting at 3 weeks of age (*Yu et al., 2006*; *Ogiwara et al., 2007*). The cellular basis for many features of DS including seizures and premature death appears to involve disinhibition, as global deletion of *Scn1a* suppresses activity of inhibitory but not excitatory neurons in the cortex and hippocampus (*Yu et al., 2006*; *Dutton et al., 2013*), and conditional deletion of *Scn1a* from forebrain inhibitory neurons results in a DS-like phenotype similar to global *Scn1a* deletion (*Cheah et al., 2012*). For these reasons, most studies have used global or inhibitory neuron-specific *Scn1a* deletions to model DS (*Catterall, 2012*), with few studies focusing on other high-priority genetic risk factors like *Scn1a* missense mutations, which represent ~40% of DS-associated mutations (*Depienne et al., 2009*; *Parihar and Ganesh, 2013*). Thus, the extent to which expression of *Scn1a* loss-of-function mutations recapitulate features of DS remains unclear. Furthermore, despite the lethality associated with *Scn1a* mutations, nothing is known regarding how loss of *Scn1a* affects brainstem respiratory centers.

The main goal of this study was to provide the first detailed characterization of breathing in a *Scn1a* missense mutation mouse model of DS. We modeled DS by expressing a loss-of-function missense mutation (A1783V) conditionally in inhibitory neurons (referred to as $Scn1a^{\Delta E26}$ mice). The A1783V variant is a DS mutation (*Marini et al., 2007*; *Lossin, 2009*; *Klassen et al., 2014*) predicted to result in loss of function by increasing Nav1.1 voltage-dependent inactivation. We found that $Scn1a^{\Delta E26}$ mice (on a 90% C57BL6/J: : 10% 129/SvJ background) exhibited spontaneous seizure activity and premature death starting at ~2 weeks of age, thus confirming this is a model of SUDEP in DS. At this same developmental time point, $Scn1a^{\Delta E26}$ mice hypoventilate, exhibit frequent apneas under baseline conditions, and show a reduced ventilatory response to $CO_2$. This respiratory phenotype is similar to what has been described DS patients (*Kim et al., 2018*). At the cellular level in a key brainstem respiratory chemoreceptor region known as the retrotrapezoid nucleus (RTN), we found that inhibitory neurons expressing the A1783V pathological variant show less spontaneous activity and a diminished ability to maintain firing during sustained depolarization. This is consistent with the possibility that the A1783V channel mutant disrupts channel expression or function by increasing voltage dependent inactivation. Also consistent with a brainstem disinhibition mechanism, we found that basal activity and $CO_2/H^+$-sensitivity of excitatory chemosensitive RTN neurons was enhanced in slices from $Scn1a^{\Delta E26}$ mice. These results show that RTN chemoreceptor function is altered in this DS model and may contribute to premature death.

## Results

### $Scn1a^{\Delta E26}$ mice have spontaneous seizures and die prematurely

To generate mice that heterologously express the *Scn1a* A1783V pathological variant conditionally in inhibitory neurons ($Scn1a^{\Delta E26}$ mice), we crossed floxed stop $Scn1a^{A1783Vfl/+}$ mice ($Scn1a^{fl/+}$) with those that express Cre recombinase targeted to *Slc32a1*, the gene that encodes the vesicular GABA transporter Vgat, to generate a $Slc32a1^{cre/+}:TdT^{+/-}$ ($Slc32a1^{cre/+}$) line (see *Figure 1—figure*

*supplement 1*). To determine whether transcript containing the A1783V variant is expressed in control or experimental animals, we isolated brainstem tissue from 13 day old pups of each genotype for subsequent cDNA amplification and sequencing. We were able to detect the expected amplicon product size of 831 base pairs in tissue from each genotype (*Figure 1Ai–Aii*). Importantly, we also were able to detect the alanine to valine single nucleotide substitution at position 1772 (analogous to position 1783 in human) in 8 of 20 (40%) samples of $Scn1a^{\Delta E26}$ tissue (*Figure 1Aiii*), indicating this missense mutation is expressed at the mRNA level. As expected, only wild type sequence was detected in $Slc32a1^{cre/+}$ and $Scn1a^{fl/+}$ control tissue (*Figure 1Aiii*), suggesting there is minimal leaky expression of A1783V in the absence of Cre.

To characterize the cellular distribution of *Scn1a* in the RTN, we prepared brainstem sections containing the RTN from $Slc32a1^{cre/+}$ and $Scn1a^{\Delta E26}$ mice (15 days old) for fluorescent in situ hybridization using probes for (1) *Scn1a*, which does not distinguish *Scn1a* channel variants; (2) *Slc32a1* gene which encodes Vgat to identify GABAergic and glycinergic inhibitory neurons; and (3) *Slc17a6* gene which encodes the vesicular glutamate transporter 2 (Vglut2) to identify excitatory glutamatergic neurons, including chemosensitive RTN neurons. We labeled all cell nuclei with DAPI. Inhibitory Vgat$^+$ cells were present in the RTN region and were in close proximity to excitatory Vglut2$^+$ neurons (i.e., putative RTN chemoreceptors). Both genotypes showed similar relative distributions of Vgat$^+$ cells ($T_{172} = 0.142$, p=0.88). We also observed numerous bright fluorescent puncta, which corresponded to *Scn1a* transcript in the soma of Vgat$^+$ cells and, to a lesser extent, in Vglut2$^+$ cells in slices from control mice ($F_{3,321} = 24.07$, p<0.0001). In slices from $Scn1a^{\Delta E26}$ mice, we found a modest reduction in *Scn1a* transcript in Vgat$^+$ but not Vglut2$^+$ cells ($F_{3,321} = 24.07$, p<0.05; see *Figure 1B–C*). Together with our sequencing data (*Figure 1Aiii*), these results suggest that the A1783V pathogenic variant is expressed by brainstem inhibitory neurons but possibly at slightly reduced levels compared to control. Therefore, cellular and behavioral phenotypes associated with $Scn1a^{\Delta E26}$ mice (see below) may involve either reduced expression, impaired channel function, or both. In a separate experiment to validate cell-type-specific Cre expression, we confirmed that all TdT$^+$ cells expressed Vgat, but not Vglut2, mRNA (not shown).

Based on previous evidence showing that heterozygous deletion mutations can give rise to severe forms of DS (*Yu et al., 2006*; *Miller et al., 2014*) and since $Scn1a^{\Delta E26}$ mice express *Scn1a* transcript, we expect $Scn1a^{\Delta E26}$ mice to exhibit a mild epilepsy-like phenotype. Contrary to this expectation, $Scn1a^{\Delta E26}$ mice show a severe SUDEP-like phenotype. $Scn1a^{\Delta E26}$ mice were born in the expected ratios, were viable, and by ~15 days postnatal, were similar in terms of body weight ($T_{46} = 1.62$, p=0.11) and temperature ($T_{26} = 0.77$, p=0.44) as their $Slc32a1^{cre/+}$ control littermates (*Figure 1D–F*). However, $Scn1a^{\Delta E26}$ pups show seizure-like behavior by ~2 weeks of age (*Table 1*). More specifically, based on the Racine seizure-behavior scoring paradigm, only 22.7% of $Slc32a1^{cre/+}$ control mice (N = 22) showed seizure-like behavior, which mainly manifested as head-bobbing (category 1). By contrast, 77.3% of $Scn1a^{\Delta E26}$ mice (N = 22) showed severe seizure behavior, including forelimb tremor (category 3), rearing alone (category 4) or in conjunction with falling over, and full-body tonic-clonic seizure (category 5). Unlike $Slc32a1^{cre/+}$ control animals, several of the mutant mice exhibited behavioral arrest that may reflect absence seizure-like activity. Furthermore, since febrile seizures are considered a hallmark of DS, we also characterized the susceptibility of $Slc32a1^{cre/+}$ control and $Scn1a^{\Delta E26}$ pups (mixed sex, 12–14 days old) to heat-induced seizures. When core body temperature was increased from 37°C to 42.5°C (0.5°C increments every 2 min) all $Scn1a^{\Delta E26}$ mice (N = 9) developed tonic-clonic seizures (category 5) at an average body temperature of 41.1 ± 0.2°C (*Table 2*). Conversely, none of the $Slc32a1^{cre/+-}$ litter mate controls (N = 10) showed seizure activity up to the cut-off temperature of 42.5°C (*Table 2*). These results differ somewhat from previous work that showed heterozygous *Scn1a* knockout mice do not develop temperature-induced seizures until ~18 days of age (*Oakley et al., 2009*). In addition to increased febrile seizure propensity, $Scn1a^{\Delta E26}$ also begin to die at ~12 days of age, reaching 100% lethality by 23 days postnatal (*Figure 1F* and *Figure 1—source data 1*). This early onset of premature death in $Scn1a^{\Delta E26}$ mice occurs ~1 week prior to mortality in heterozygous *Scn1a* knockout mice on a pure C57BL/6J background (*Yu et al., 2006*; *Catterall, 2012*), suggesting $Scn1a^{\Delta E26}$ mice (90% C57BL/6J background) have a particularly severe phenotype.

To determine whether $Scn1a^{\Delta E26}$ mice exhibit abnormal brain activity, we made video electrocorticogram (ECoG) recordings from $Slc32a1^{cre/+}$ control and $Scn1a^{\Delta E26}$ mice. We allowed mice 12 hr to recover after implanting them with the ECoG head stage. We continuously recorded animal behavior

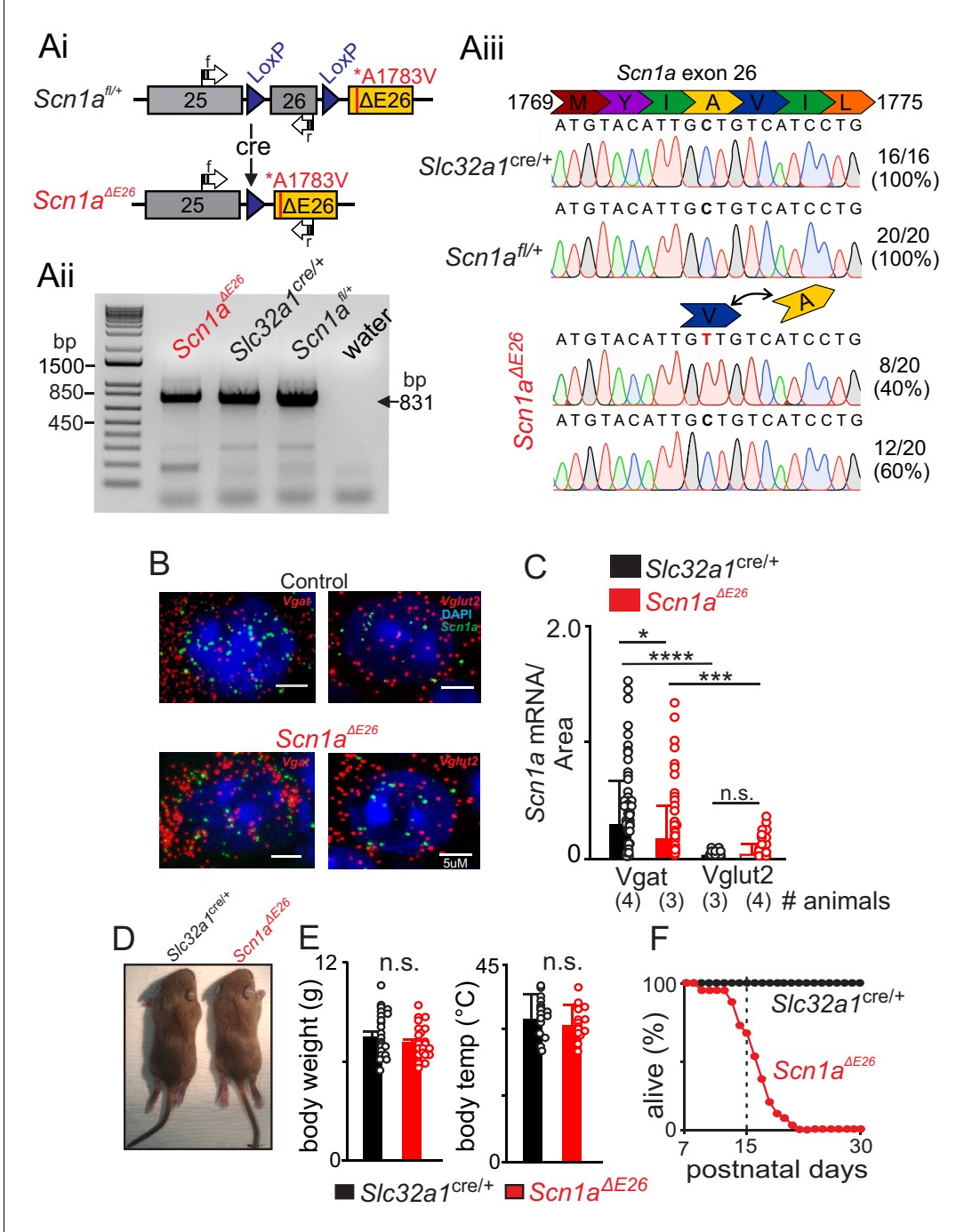

**Figure 1.** Conditional expression of *Scn1a* A1783V in inhibitory neurons results in premature death. (**A**) Construct design and validation of A1783V variant transcript expression. Note that this line was developed by Ana Mingorance (Chief Development Officer of the Loulou Foundation) and is available at JAX (sock # 026133). Ai, schematic shows loxP sites flanking wild type exon 26 followed by an edited version of exon 26 that contains the human A1783V pathological variant (ΔE26). When Cre recombinase is expressed, wild type exon 26 is removed, thus allowing transcription of ΔE26. Aii, Agarose gel shows detectable levels of *Scn1a* transcript (expected size of 831 bp) in brainstem tissue isolated from each genotype (primers span between exon 25 and 26, including residue 1783 of exon 26). Water was used as a no template negative control. Aiii, PCR products were sequenced to confirm that transcript containing A1783V is detectable in 40% of samples from *Scn1a*^ΔE26 tissue but was not detectable in samples from *Slc32a1*^cre/+ and *Scn1a*^fl/+ control tissue. (**B–C**), fluorescent in situ hybridization (RNAScope) was performed to characterize expression of *Scn1a* transcript in inhibitory (*Slc32a1*^+, *Slc32a1*) and glutamatergic (Vglut2^+, *Slc17a6*) neurons in the RTN region in brainstem sections from control and *Scn1a*^ΔE26 mice. (**B**) brainstem sections from *Slc32a1*^cre/+ and *Scn1a*^ΔE26 mice containing the RTN show *Scn1a* labeling (green puncta) of both Vgat^+ and Vglut2 +neurons. (**C**), summary data show *Scn1a* transcript expression (normalized to cell size) in Vgat^+ and Vglut2 +RTN neurons from each genotype; channel transcript

*Figure 1 continued on next page*

*Figure 1 continued*

was reduced in Vgat[+] cells from *Scn1a*[ΔE26] mice (0.43 ± 0.7 mRNA/area, n = 94 cells) compared to control (0.73 ± 0.9 mRNA/area, n = 82 cells) (p<0.05), whereas Vglut2 +cells showed low channel transcript across both genotypes. (**D–E**) *Scn1a*[ΔE26] mice did not show any obvious differences gross morphology (**A**) body weight (**D**) or temperature (**E**) compared to age-matched litter mate control mice. (**F**) (*Figure 1—source data 1*), survival curve shows that control mice (n = 57) survive to adulthood (30 days postnatal) while *Scn1a*[ΔE26] mice (n = 41) die prematurely starting at 9 days postnatal and reaching 100% lethality by 25 days ($\chi^2$ = 63.9, p<0.0001). These results were compared using a two-way ANOVA and Sidak multiple comparison test. *, p<0.05; ***p<0.001; ****p<0.0001.

DOI: https://doi.org/10.7554/eLife.43387.002

The following source data and figure supplement are available for figure 1:

**Source data 1.** Survival curves for *Slc32a1*[cre/+] and *Scn1a*[ΔE26] mice.

DOI: https://doi.org/10.7554/eLife.43387.004

**Figure supplement 1.** Breeding strategy to generate mice that heterologously express the *Scn1a* A1783V pathological variant conditionally in inhibitory neurons (Scn1a[ΔE26] mice).

DOI: https://doi.org/10.7554/eLife.43387.003

and ECoG activity over a two-hour period, between the hours of 9:00 AM – 2:00 PM. Consistent with frequent polyspike activity observed in the ECoG recordings of DS patients (*Bender et al., 2012*), *Scn1a*[ΔE26] mice showed large amplitude (at least twice baseline) polyspike activity that lasted for an average duration of 15.6 ± 0.8 s (*Figure 2A–Bi*). These events always occurred in conjunction with seizure activity (category 4–5) but were frequently preceded by brief behavioral arrest. The duration of these polyspike events were considerably shorter than spike wave discharges associated with absence epilepsy that typically last for >1 s (*Letts et al., 2014*). Conversely, *Slc32a1*[cre/+] litter-mate control animals showed minimal large amplitude spike activity; any detectable events were of a short duration 7.55 ± 0.6 s (*Figure 2A–Bi*) and occurred when the animal was exhibiting exploratory behavior and less obvious seizure like activity (category 0–2) or freezing behavior. Based on this, we define epileptic spike activity for this model as abrupt onset polyspiking events with greater than twice baseline amplitude, minimum duration of 14 ms, and that occur in conjunction with seizure activity (category 3–5). Based on this criteria, *Scn1a*[ΔE26] and *Slc32a1*[cre/+] control mice show epileptic spike activity with a frequency 9.167 ± 3.9 events/2 hr and of 0.25 ± 0.1 events/2 hr, respectively ($T_5$ = 2.28, p<0.05) (*Figure 2Bii*). Note that three *Slc32a1*[cre/+] animals showed a polyspike event that lasted longer than 14 ms and occurred with noticeable forelimb shaking and so were included in our analysis (*Figure 2Biii*). Power spectral analysis of poly-spike events in *Slc32a1*[cre/+] and epileptic events in *Scn1a*[ΔE26] mice show that epileptic events were composed of high alpha and beta frequencies ($F_{4, 840}$ = 5.605, p<0.001) (*Figure 2C–D* and *Figure 2—source data 1*). These results suggest *Scn1a*[ΔE26] mice are phenotypically similar to global and inhibitory neuron-specific *Scn1a* haploinsufficient models of DS (*Yu et al., 2006*; *Kalume et al., 2013*; *Kim et al., 2018*), but with an accelerated time course for manifestation of pathological features including spontaneous and heat-induced seizures as well as premature death. Based on the above results, we consider the *Scn1a*[ΔE26] mouse model to be useful for dissecting the mechanisms that underlie respiratory failure in DS.

## *Scn1a*[ΔE26] mice hypoventilate under baseline conditions and have a reduced $CO_2/H^+$ ventilatory response

Recent evidence (*Kim et al., 2018*) showed that DS patients have post-ictal respiratory abnormalities, including hypoventilation, apnea and impaired $CO_2$ chemoreception. These symptoms can last for several hours after seizure, which indicates that respiratory problems contribute to SUDEP in DS. Therefore, to determine whether *Scn1a*[ΔE26] mice exhibit respiratory problems, we used whole-body

**Table 1.** Behavioral Assessment of Seizure activity.

| Racine score | N | 0 | 1 | 2 | 3 | 4 | 5 | Behavioral arrest |
|---|---|---|---|---|---|---|---|---|
| *Slc32a1*[cre/+] | 22 | 17 | 5 | 0 | 0 | 0 | 0 | 0 |
| *Scn1a*[ΔE26] | 22 | 0 | 3 | 3 | 6 | 2 | 3 | 5 |

DOI: https://doi.org/10.7554/eLife.43387.005

**Table 2.** Febrile seizure propensity.

| Genotype | N | Weight | Induced seizure (%) |
|---|---|---|---|
| $Slc32a1^{cre/+}$ | 10 | $6.65 \pm 0.3$ | 0 |
| $Scn1a^{\Delta E26}$ | 9 | $6.71 \pm 0.3$ | 100**** |

****Fisher's exact test p<0.000

DOI: https://doi.org/10.7554/eLife.43387.006

The following source data is available for Table 2:

Source data 1. Febrile seizure propensity.

DOI: https://doi.org/10.7554/eLife.43387.007

plethysmography to measure baseline breathing and the ventilatory response to $CO_2$ in 15-day-old control and $Scn1a^{\Delta E26}$ mice. Note that there were no measureable differences in respiratory activity between $Slc32a1^{cre/+}$ (n = 16) and $Scn1a^{fl/+}$ (N = 5) mice (p=0.267). Therefore, these genotypes were pooled as controls for this set of experiments. We found that compared to control animals, $Scn1a^{\Delta E26}$ mice show diminished respiratory output under room air conditions. Specifically, $Scn1a^{\Delta E26}$ exhibit suppressed frequency (256 ± 11 bpm for control compared to 211 ± 16 bpm for $Scn1a^{\Delta E26}$; $T_{33}$ = 2.43; p<0.05); tidal volume (14.8 ± 2.0 µl/g for control compared to 8.9 ± 1.9 µl/g for $Scn1a^{\Delta E26}$, $T_{33}$ = 2.02, p<0.05); and minute ventilation (3.6 ± 0.5 µl/min/g for control compared to 2.7 ± 0.5 µl/min/g for $Scn1a^{\Delta E26}$; $T_{33}$ = 2.01, p<0.05; *Figure 3A–D*). Although both control and $Scn1a^{\Delta E26}$ mice exhibit apneic events at similar frequencies (0.23 ± 0.1/min for control and 0.11 ± 0.04/min for $Scn1a^{\Delta E26}$; p=0.6), the duration of these events were longer in $Scn1a^{\Delta E26}$ mice (*Figure 3E*; 1,104 ± 58.6 ms for controls versus 1,350 ± 99.2 ms $Scn1a^{\Delta E26}$; $T_{51}$ = 2.135; p<0.05). We also found that $Scn1a\Delta^{E26}$ mice had a diminished capacity to increase respiratory frequency in response to graded increases in $CO_2$ (*Figure 3F*). Specifically, respiratory frequency in 7% $CO_2$ (balance $O_2$) was higher in controls (363.1 ± 7.7 bpm; N = 22) versus $Scn1a^{\Delta E26}$ (300.7 ± 17.4 bpm; N = 17; $F_{1,37}$ = 5.69, p<0.05). Although tidal volume responses to $CO_2$/$H^+$ are similar between genotypes (p=0.47), total respiratory output, as measured by minute ventilation— the product of respiratory frequency and tidal volume—was diminished in $Scn1a^{\Delta E26}$ mice compared to controls ($F_{3,111}$ = 3.167, p<0.05; *Figure 3G–H* and *Figure 3—source data 1*). Specifically, increasing inspired $CO_2$ from 0% to 3% increased minute ventilation in control mice by 3.3 ± 0.5 µl/min/g (p<0.0001). These same conditions, however, led to a much smaller and non-significant increase in minute ventilation among $Scn1a^{\Delta E26}$ mice (increase of 1.5 ± 0.5 µl/min/g; p=0.07). These results show that $Scn1a^{\Delta E26}$ mice exhibit a respiratory phenotype similar to that observed in DS patients, and further supports the possibility that respiratory problems may contribute to mortality in this DS model.

## Disinhibition and altered RTN chemoreception in $Scn1a^{\Delta E26}$ mice

The RTN regulates several aspects of breathing including chemoreception (*Guyenet and Bayliss, 2015*) and since the ventilatory response to $CO_2$ is disrupted in DS patients (*Kim et al., 2018*), we wanted to determine whether activity of chemosensitive RTN neurons are disrupted in $Scn1a^{\Delta E26}$ mice. Furthermore, evidence also suggests that inhibitory neurons in the RTN region contribute to respiratory drive (*Ott et al., 2011*). Therefore, we first sought to determine whether loss of $Scn1a$ function in inhibitory neurons decreases inhibitory neuron activity and disinhibits excitatory, chemosensitive, neurons. Inhibitory neurons were identified by Cre-dependent TdT labeling of Vgat$^+$ cells in both control and $Scn1a^{\Delta E26}$ lines. Consistent with other $Scn1a$ knockout (*Tai et al., 2014*) or missense knockin (*Ogiwara et al., 2007*; *Mashimo et al., 2010*; *Hedrich et al., 2014*) DS models, we found that loss of $Scn1a$ function in inhibitory neurons suppressed inhibitory neural activity. For example, whole-cell current-clamp recordings from inhibitory neurons in the RTN region in slices from $Slc32a1^{cre/+}$ or $Scn1a^{\Delta E26}$ mice show that inhibitory neurons from $Scn1a^{\Delta E26}$ mice have lower basal activity than those of $Slc32a1^{cre/+}$ control mice (14.39 ± 1.5 Hz for $Slc32a1^{cre/+}$ vs. 9.902 ± 0.64 Hz for $Scn1a^{\Delta E26}$; $T_{60}$ = 2.97, p<0.01; *Figure 4A–B*). Furthermore, $Scn1a^{\Delta E26}$ inhibitory neurons fired fewer action potentials in response to depolarizing current steps (0–300 pA; Δ 20 pA) from a holding potential of −80 mV. This activity deficit became more pronounced during large (200–300 pA) sustained (1,000 ms) current injections where inhibitory neurons from $Scn1a^{\Delta E26}$ mice showed

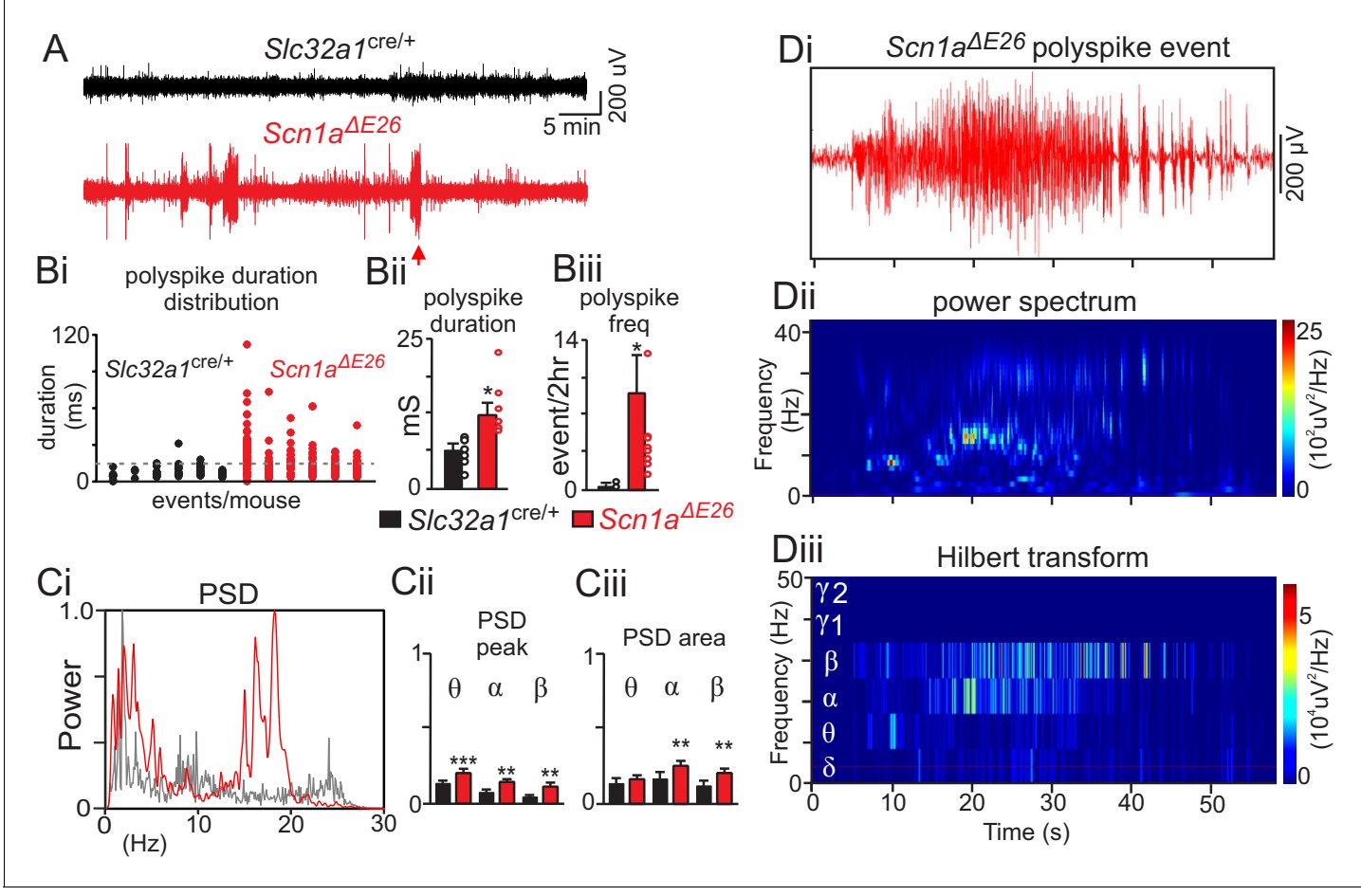

**Figure 2.** $Scn1a^{\Delta E26}$ exhibit frequent spontaneous seizures. (**A**) traces of raw EcoG activity show that $Scn1a^{\Delta E26}$ mice but not $Slc32a1^{cre/+}$ mice exhibit frequent spontaneous burst of high amplitude poly-spike activity. The arrow identifies a typical seizure-like poly-spike event that was analyzed further by power spectral analysis in panel D. Polyspike events with a minimum duration of 14 ms were accompanied by seizure-like behavior and so were considered epileptic activity. (**B**), $Scn1a^{\Delta E26}$ mice showed more frequent epileptic poly-spike bursts of activity (Bi, dotted line designates duration threshold for epileptic activity); poly-spike bursts (>14 ms) occurred more frequently in $Scn1a^{\Delta E26}$ mice (Bi-Bii) (control 0.13 ± 0.1 events/2 hr, n = 6; $Scn1a^{\Delta E26}$ 0.37 ± 0.05 events/2 hr, n = 6; $T_{10}$ = 3.009, p<0.01) and lasted for a longer duration (Biii) (control 7.6 ± 0.6 ms, n = 6; $Scn1a^{\Delta E26}$ 15.6 ± 0.8 ms, n = 6, $T_{10}$ = 2.268, p<0.05) compared to control animals. Ci, representative power spectrum density (PSD) plots of spontaneous poly-spike burst events show typical strong activity in the theta-, alpha and beta frequency range in $Scn1a^{\Delta E26}$ but not control mice. Cii-Ciii (*Figure 2—source data 1*), summary data (normalized to the maximum value at each event) show PSD peak (Cii) and PSD area under the curve (Ciii) of each frequency range for each genotype. Note that poly-spike burst events measured in $Scn1a^{\Delta E26}$ mice show increased activity in the theta, alpha and beta range. Di-iii, poly-spike burst events recorded from a $Scn1a^{\Delta E26}$ mouse (arrow in panel A) plotted on an expanded time scale (Di) and corresponding time frequency distribution (Dii) and deconstructed spectrum into its various frequency domains (Diii). These results were compared using a two-way ANOVA and the Sidak multiple comparison test. *, p<0.05; **, p<0.01; ***p<0.001.

DOI: https://doi.org/10.7554/eLife.43387.008

The following source data is available for figure 2:

**Source data 1.** Plots of PSD peak and PSD area of poly-spike burst events measured in $Slc32a1^{cre/+}$ and $Scn1a^{\Delta E26}$ mice.

DOI: https://doi.org/10.7554/eLife.43387.009

pronounced spike amplitude and frequency decrement (*Figure 4A,D* and *Figure 4—source data 1*). That is, the number of spikes elicited by a + 300 pA current step (1,000 ms) was 53.7 ± 11 for $Slc32a1^{cre/+}$ controls (N = 13) compared to 13.9 ± 6.4 for $Scn1a^{\Delta E26}$ (N = 20; $F_{15,465}$ = 9.536; p<0.0001). We also found that inhibitory neurons from each genotype had similar input resistance (517.6 ± 82.2 MΩ for $Slc32a1^{cre/+}$ control vs. 519.2 ± 38.9 MΩ for $Scn1a^{\Delta E26}$; $T_{25}$ = 0.02; p=0.3; *Figure 4C*). These results show that inhibitory neurons in slices from $Scn1a^{\Delta E26}$ mice have diminished spontaneous activity and a reduced ability to respond to a range of excitatory inputs.

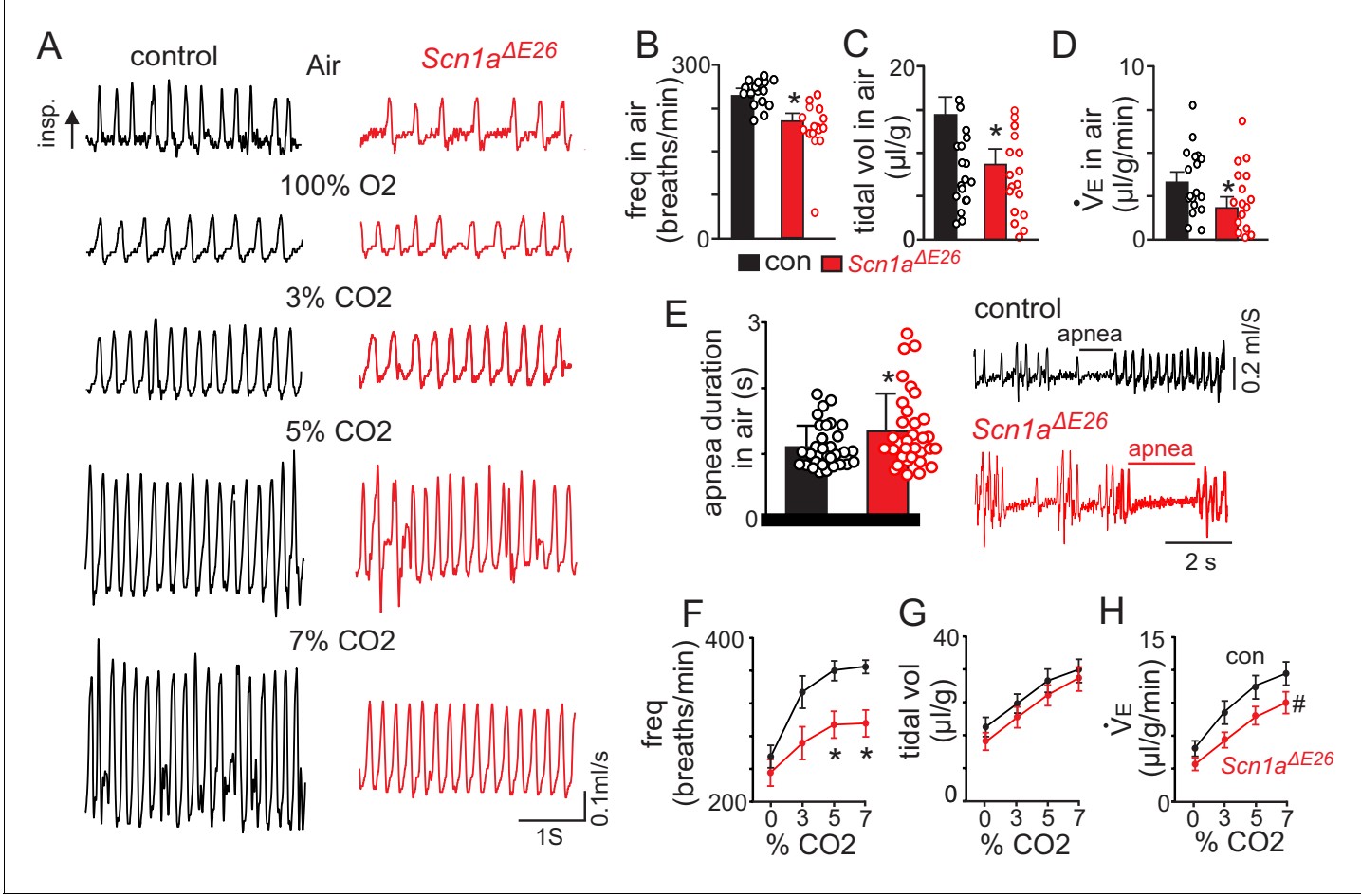

**Figure 3.** $Scn1a^{\Delta E26}$ mice show reduced respiratory output under control conditions and during exposure to high $CO_2$. For these experiments $Scn1a^{fl/+}$ and $Slc32a1^{cre/+}$ were used as control. (**A**) traces of respiratory activity from a control and $Scn1a^{\Delta E26}$ mouse during exposure to room air, 100% $O_2$ and 3–7% $CO_2$ (balance $O_2$). (**B–D**), summary data (n = 22 control; n = 17 $Scn1a^{\Delta E26}$) show respiratory frequency (**B**), tidal volume (**C**) and minute ventilation (**D**) are reduced in $Scn1a^{\Delta E26}$ mice compared to control under room air conditions. (**E**), traces of respiratory activity (left) and summary data (right) show that under room air conditions both control and $Scn1a^{\Delta E26}$ mice exhibit periods of apnea; the frequency of these events were similar between genotypes, however, they lasted for a longer duration in $Scn1a^{\Delta E26}$ mice compared to control. F-H (*Figure 3—source data 1*), summary data shows the respiratory frequency (**F**), tidal volume (**G**) and minute ventilation response of control and $Scn1a^{\Delta E26}$ mice to graded increases in $CO_2$ (balance $O_2$). $Scn1a^{\Delta E26}$ mice showed a blunted respiratory frequency to 5% and 7% $CO_2$ which resulted in a diminished $CO_2/H^+$-dependent increase in minute ventilation. These results were compared using either unpaired t test (panels B-E) or two-way ANOVA followed by the Holm-Sidak multiple comparison test (panels F-H). *, difference between means p<0.05, #, different interaction factor, p<0.05.

DOI: https://doi.org/10.7554/eLife.43387.010

The following source data is available for figure 3:

**Source data 1.** Summary data showing respiratory frequency, tidal volume and minute ventilatory responses to $CO_2$ in control and $Scn1a^{\Delta E26}$ mice.

DOI: https://doi.org/10.7554/eLife.43387.011

The A1783V pathological variant is located in the S6 segment of domain 4 (*Marini et al., 2007*; *Lossin, 2009*), a region thought to regulate voltage-dependent inactivation (*Catterall, 2000*). Based on our evidence that A1783V is expressed in tissue form $Scn1a^{\Delta E26}$ mice (*Figure 1Aiii*) and since inhibitory neurons from these animals show reduced excitability (*Figure 4A–B*), we hypothesized that the $Scn1a$ A1783V variant results in loss of function, in part, by causing Nav1.1 channels to inactivate at more negative voltages. Consistent with this hypothesis, when examining spontaneous action potentials (as measured under resting conditions with a 0 pA holding current) in inhibitory neurons in slices of $Scn1a^{\Delta E26}$ and $Slc32a1^{cre/+}$ control mice, the latter showed a higher amplitude (73.5 ± 1.9 mV) than $Scn1a^{\Delta E26}$ (61.9 ± 2.6 mV; $F_{1,95}$ = 9.931, p<0.001). The maximum rate of depolarization was higher for $Slc32a1^{cre/+}$ controls (134.4 ± 5.0 mV/mS) compared to $Scn1a^{\Delta E26}$

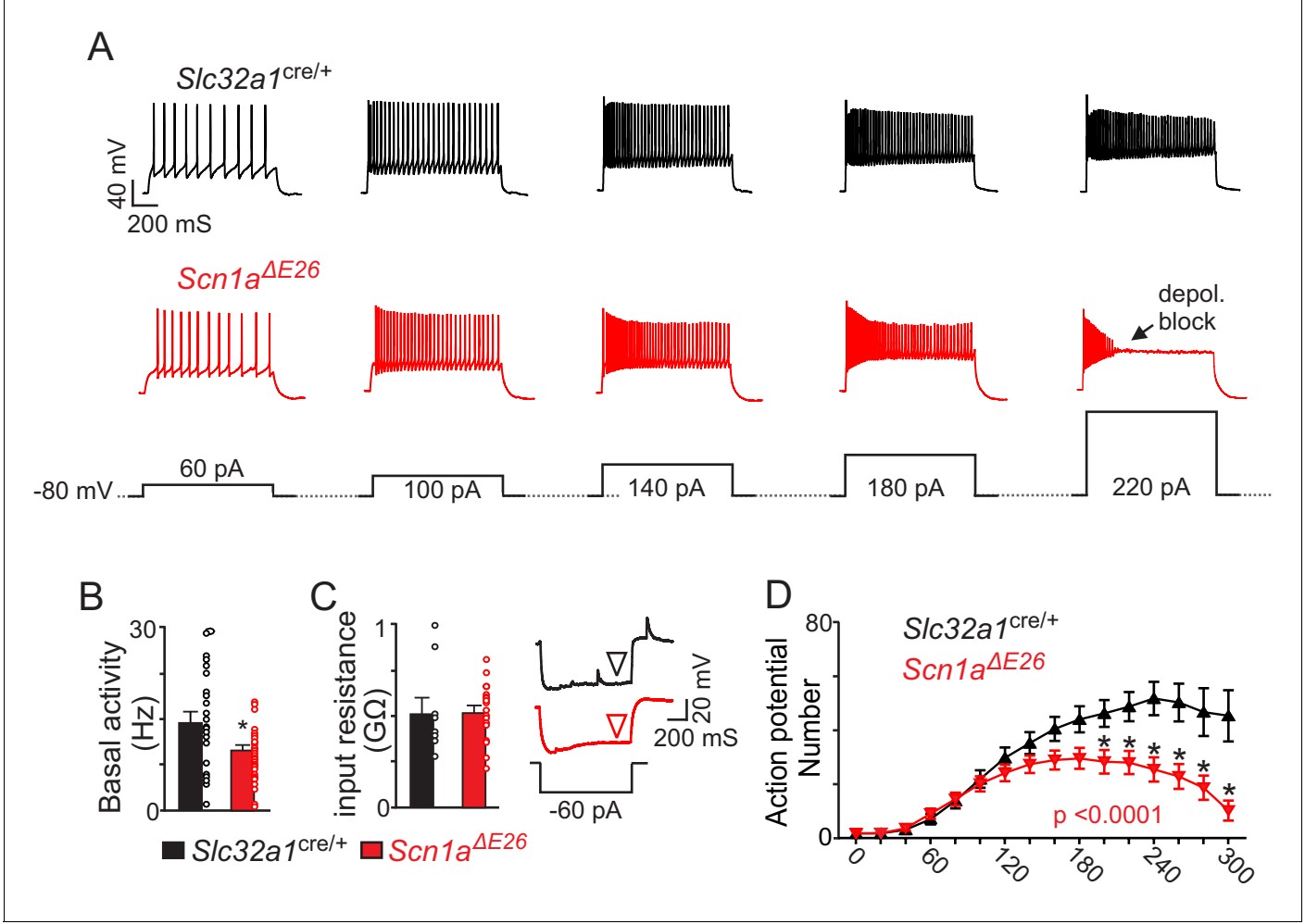

**Figure 4.** Brainstem inhibitory neurons in slices from $Scn1a^{\Delta E26}$ show diminished basal activity and repetitive firing behavior during sustained depolarization. (A) segments of membrane potential from inhibitory neurons in the RTN region in slices from control and $Scn1a^{\Delta E26}$ mice during depolarizing current injections (40 to 220 pA; 1 s duration) from a membrane potential of –80 mV. (B) summary data shows inhibitory neurons in slices from $Scn1a^{\Delta E26}$ mice (n = 36) are less active under resting conditions (0 pA holding current) compared to inhibitory neurons in slices form $Slc32a1^{cre/+}$ control mice (n = 26 cells). (C), summary data and representative voltage responses to a −60 pA current injection show that inhibitory neurons from each genotype had similar input resistance. (D) (*Figure 4—source data 1*), input-output relationship show that inhibitory neurons from $Scn1a^{\Delta E26}$ mice generate fewer action potentials in response to moderate depolarizing current injections (1 s duration) and at more positive steps go into depolarizing block. Results were compared using t-test (B–C) and two-way ANOVA and Sidak multiple comparison test (D). *, p<0.05; **, p<0.01; ***, p<0.001.

DOI: https://doi.org/10.7554/eLife.43387.012

The following source data is available for figure 4:

**Source data 1.** Evoked firing responses of inhibitory neurons in slices from $Slc32a1^{cre/+}$ and $Scn1a^{\Delta E26}$ mice.

DOI: https://doi.org/10.7554/eLife.43387.013

(89.4 ± 5.3 mV/mS; $F_{1,96}$ = 35.2, p<0.0001; see *Figure 5A,D–F* and *Figure 5—source data 1*). Action potential threshold was also higher in inhibitory neurons in slices from $Scn1a^{\Delta E26}$ (−29.2 ± 0.9 mV) compared to $Slc32a1^{cre/+}$ controls (−32.4 ± 0.6 mV; $F_{1,95}$ = 7.403, p<0.05; *Figure 5A,F*).

Next, we characterized the properties of the first action potential elicited after holding cells at potentials that either remove or enhance Nav1.1 channel inactivation. We found that differences in action potential waveform properties between genotypes were minimized when cells are held at a negative voltage to remove $Na^+$ channel inactivation. For example, holding inhibitory neurons in slices from $Scn1a^{\Delta E26}$ mice at a negative pre-potential by injecting a hyperpolarizing current (−100 pA; 1,000 ms) increased action potential amplitude (78.01 ± 2.0 mV; $F_{1,95}$ = 9.931, p<0.0001) to an amount similar to spikes from $Slc32a1^{cre/+}$ control cells (81.05 ± 1.2 mV; p=0.83) (*Figure 5B,E*).

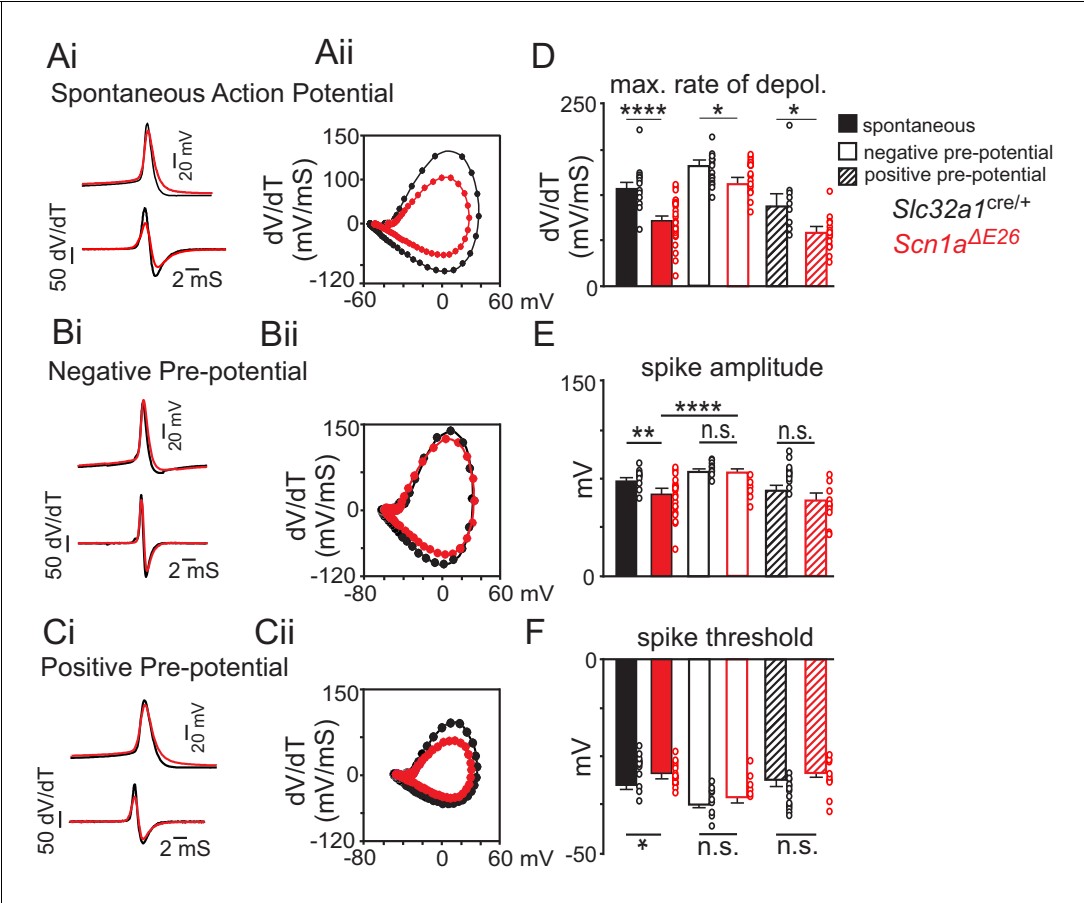

**Figure 5.** The *Scn1a* A1783V pathological variant may result in loss of channel function by increased voltage-dependent inactivation. (A) average spontaneous action (*Slc32a1*^cre/+ control n = 24 spikes, *Scn1a*^ΔE26 n = 29 spikes (top) and first time derivative of average action potentials (bottom) recorded from inhibitory neurons in slices from control and *Scn1a*^ΔE26 mice (Ai) and corresponding phase plot (Aii) (dV/dt; Y-axis vs mV; X-axis) of the traces in panel Ai show that cells expressing *Scn1a* A1783V depolarize slower compared to control cells (***Figure 5—source data 1***). (B), average first action potential following a hyperpolarizing pre-potential (−100 pA; 1 s) (*Slc32a1*^cre/+ control n = 13 spikes, *Scn1a*^ΔE26 n = 16 spikes) (top) and first time derivative of average action potentials (bottom) recorded from inhibitory neurons in slices from control and *Scn1a*^ΔE26 mice (Bi) and corresponding phase plot (Bii) of traces in panel Bi show that holding cells at a negative pre-potential to remove sodium channel inactivation improved the depolarization kinetics of subsequent spikes (***Figure 5—source data 1***). (C), average first action potential following a depolarizing pre-potential (+180 pA; 1 s) (control n = 9 spikes, *Scn1a*^ΔE26 n = 11 spikes (top) and first time derivative of average action potentials (bottom) recorded from inhibitory neurons in slices from control and *Scn1a*^ΔE26 mice (Ci) and corresponding phase plot (Cii) of traces in panel Ci show that holding cells at a depolarized pre-potential to increase sodium channel inactivation diminished genotype differences in action potential kinetics (***Figure 5—source data 1***). (D–F), summary data showing the maximum rate of depolarization (D), action potential amplitude (E) and action potential threshold (F) of spontaneous action potentials and first spikes following positive or negative pre-potentials recorded in slices from control and *Scn1a*^ΔE26 mice. Results were compared by two-way ANOVA and the Sidak multiple comparison test.. *, p<0.05; **, p<0.01; ***, p<0.001; ****, p<0.0001.
DOI: https://doi.org/10.7554/eLife.43387.014

The following source data is available for figure 5:

**Source data 1.** Phase plots of spontaneous action potentials and the first spike elicited following positive or negative pre-potentials in slices from *Slc32a1*^cre/+ and *Scn1a*^ΔE26 mice.
DOI: https://doi.org/10.7554/eLife.43387.015

Under these conditions, the maximum rate of depolarization also increased 140.3 ± 5.441 mV/ms ($F_{1, 96}$ = 35.21, p<0.0001) (***Figure 5B,D*** and ***Figure 5—source data 1***); this rate was similar to that measured in spontaneous spikes from *Slc32a1*^cre/+ control animals (p>0.99) but slower than spikes from *Slc32a1*^cre/+ control cells following a negative pre-potential (166.6 ± 7.3 mV/mS, $F_{1,96}$ = 35.21, p<0.05). Holding inhibitory neurons in slices from *Scn1a*^ΔE26 mice at a negative pre-potential also lowered the threshold for spike initiation (−35.68 ± 0.7 mV; $F_{1, 95}$ = 7.403, p<0.001) to a level similar to *Slc32a1*^cre/+ control cells (−37.2 ± 1.2 mV; $F_{1, 95}$ = 7.403, p=0.06) (***Figure 5B,F***). We also found

that delivering a + 180 pA current for 1,000 ms to enhance $Na^+$ channel inactivation in inhibitory neurons in slices from $Slc32a1^{cre/+}$ control mice resulted in similar action potential amplitude ($F_{1, 95}$ = 9.931, p>0.99), rate of depolarization ($F_{1, 96}$ = 35.21, p=0.58) and spike threshold ($F_{1, 95}$ = 7.403, p=0.97) as spikes measured in inhibitory neurons from $Scn1a^{\Delta E26}$ slices under resting conditions (holding current = 0 pA) (*Figure 5C,D–F* and *Figure 5—source data 1*). Although it is possible that diminished expression of channel containing A1783V (*Figure 1C*) may also contribute to this electro-physiological phenotype; since transcript containing A1783V appears to be abundantly expressed (*Figure 1Aiii*), we speculate that loss of *Scn1a* function involves enhanced Nav1.1 inactivation.

Based on previous evidence suggests inhibitory neurons in the RTN region can regulate activity of chemosensitive neurons (*Ott et al., 2011*), we predict that loss of inhibitory tone by expression of *Scn1a* A1783V would enhance basal activity and $CO_2/H^+$ sensitivity of glutamatergic chemosensitive neurons. To test this, we characterized the firing activity of chemosensitive RTN neurons in slices from $Slc32a1^{cre/+}$ control and $Scn1a^{\Delta E26}$ mice during exposure to $CO_2$ levels ranging from 3% to 10%. We initially identified chemosensitive RTN neurons in each genotype by their firing response to $CO_2$. We considered neurons that are spontaneously active in 5% $CO_2$ and responded to 10% $CO_2$ with at least a 1.0 Hz increase in firing rate to be chemosensitive. Chemosensitive RTN neurons also have been shown to express the transcription factor Phox2b; therefore, at the end of each experiment, we filled all recorded cells with Lucifer yellow for later immunohistochemical confirmation of Phox2b expression. Chemosensitive RTN neurons in slices from $Slc32a1^{cre/+}$ control mice had an average basal activity of $1.3 \pm 0.4$ Hz under control conditions (5% $CO_2$; pH 7.3). These cells were strongly inhibited by decreasing $CO_2$ to 3% (pHo = 7.6) ($1.02 \pm 0.3$ Hz) and showed a linear firing increase in response to 7% (pHo = 7.2) ($2.4 \pm 0.5$ Hz) and 10% $CO_2$ (pHo = 7.0) ($2.8 \pm 0.4$ Hz) (*Figure 6A,B–C*). This $CO_2$ response profile is consistent with type I chemoreceptors ($pH_{50}$ = 7.3), which were described previously in a Phox2b mouse reporter line (*Lazarenko et al., 2009*). Consistent with our hypothesis, chemosensitive RTN neurons in slices from $Scn1a^{\Delta E26}$ mice were more active under control conditions (5% $CO_2$) ($2.4 \pm 0.35$ Hz) (*Figure 6C*) ($T_{21}$ = 2.223, p<0.05) and showed an enhanced firing response to high $CO_2/H^+$ (*Figure 6D* and *Figure 6—source data 1*) (slope: $0.3 \pm 0.01$ $Slc32a1^{cre/+}$ vs. $0.37 \pm 0.01$ $Scn1a^{\Delta E26}$, F1,4 = 8.04, p<0.05). These results show that loss of *Scn1a* function in inhibitory neurons disrupts activity of RTN chemoreceptors.

## Discussion

Epilepsy patients have a 20-fold higher mortality rate than the general population (*Massey et al., 2014*). The most common cause of death for this patient population is SUDEP, a leading cause of which is respiratory failure (*Surges et al., 2009*; *Ryvlin et al., 2013*; *Kennedy and Seyal, 2015*; *Dlouhy et al., 2016*). However, mechanisms underlying respiratory dysfunction in epilepsy and SUDEP are largely unknown. This is particularly true in the context of DS, where patients have an exceedingly high mortality rate and commonly exhibit life-threatening respiratory problems (*Kim et al., 2018*), yet little is known regarding how loss of *Scn1a* function impacts brainstem respiratory centers. The results presented here address this knowledge gap by showing that expression of the a DS-associated *Scn1a* variant A1783V in inhibitory neurons resulted in both spontaneous (*Figure 2*) and heat-induced seizures (*Table 2*) as well as pre-mature death (*Figure 1E*). Moreover, this mouse model presents with a respiratory phenotype reminiscent of that exhibited by DS patients (*Figure 3*). Perhaps not surprising, we found that loss of *Scn1a* function in inhibitory neurons in the RTN diminished activity in a cell-autonomous manner (*Figures 4–5*) but, importantly, also enhanced baseline activity and $CO_2/H^+$ sensitivity of glutamatergic chemosensitive neurons (*Figure 6*). These results suggest that disruption of *Scn1a* in inhibitory neurons can alter normal activity of brainstem respiratory centers and so may contribute to pathological features of DS including disordered breathing associated with SUDEP.

By ~2 weeks of age, $Scn1a^{\Delta E26}$ mice show respiratory abnormalities characterized by hypoventilation, increased apneas and diminished ventilatory response to $CO_2/H^+$ (*Figure 3*). This phenotype is similar to that observed in DS patients (*Kim et al., 2018*). Furthermore, these breathing problems occurred in conjunction with a marked increase in mortality (*Figure 1F* and *Figure 1—source data 1*), thus correlatively supporting the possibility that respiratory dysfunction contributes to premature death in DS. Although mechanisms contributing to respiratory dysfunction in DS are unknown, previous work showed that loss of *Scn1a* from inhibitory neurons in the forebrain, but not the brainstem

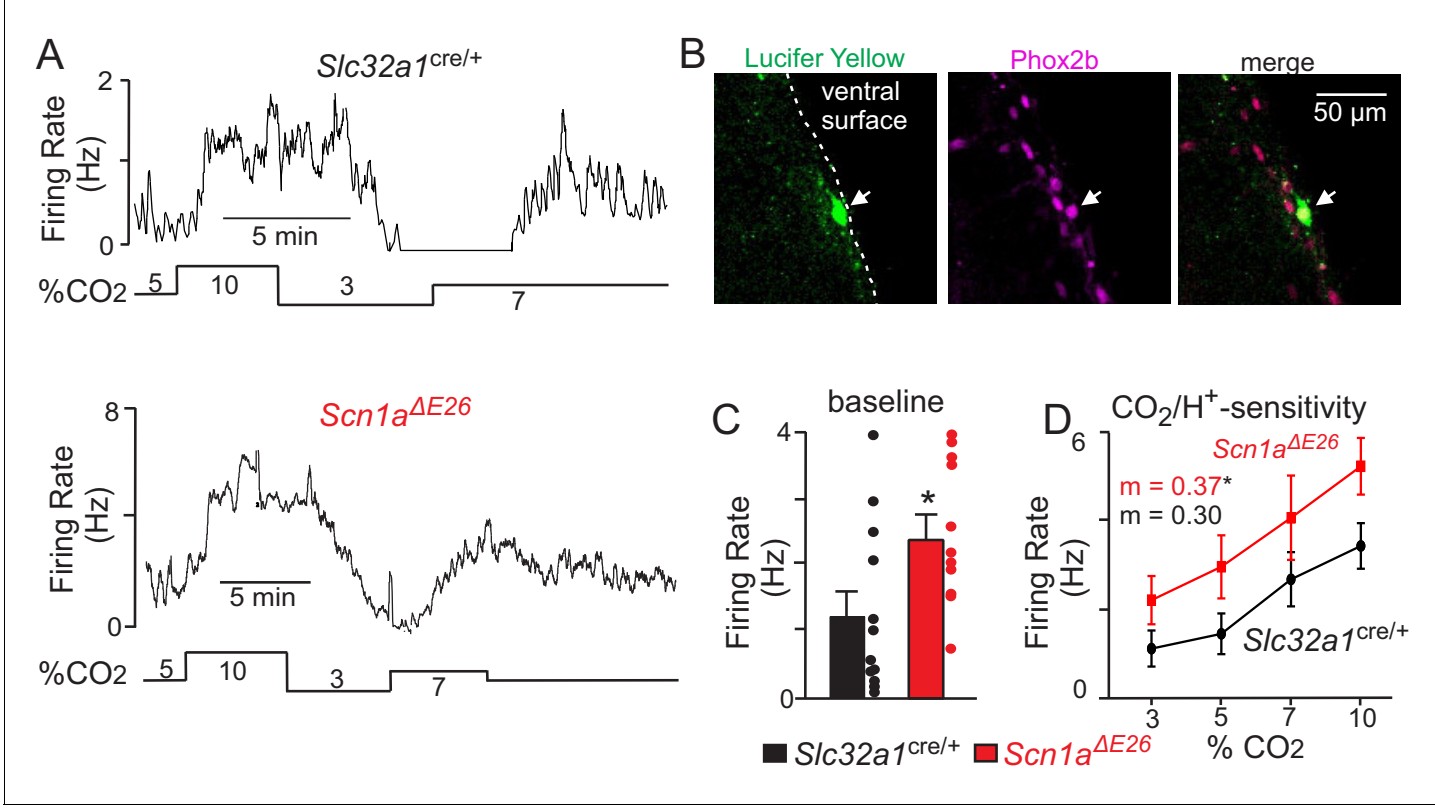

**Figure 6.** Chemosensitive RTN neurons in slices from $Scn1a^{\Delta E26}$ mice are hyper-excitable. (A) firing rate traces from chemosensitive neurons in slices from control (top) and $Scn1a^{\Delta E26}$ mice (bottom) show that neurons from both genotypes respond to changes in $CO_2/H^+$; RTN neurons are spontaneously active under control conditions (5% $CO_2$; pHo 7.3) and respond to 7% $CO_2$ (pHo 7.2) and 10% $CO_2$ (pHo 7.0) with a linear increase in activity, whereas exposure to 3% $CO_2$ (pHo 7.6) decreases neural activity. However, basal activity and $CO_2/H^+$-dependent output of RTN chemoreceptors from $Scn1a^{\Delta E26}$ tissue is enhanced compared to RTN neurons in slices from $Slc32a1^{cre/+}$ control mice. (B) double-immunolabeling shows that a Lucifer Yellow-filled $CO_2/H^+$-sensitive RTN neuron (green) is immunoreactive for phox2b (magenta), the merged image is shown to the right. We confirmed that all $CO_2/H^+$-sensitive neurons ($Slc32a1^{cre/+}$ control n = 12; $Scn1a^{\Delta E26}$ n = 11) included in this study were phox2b-positive. (C–D) (*Figure 6—source data 1*), summary data shows that RTN chemoreceptors in slices from $Scn1a^{\Delta E26}$ mice have higher basal activity (C) and enhanced $CO_2/H^+$-dependent output between 3–10% $CO_2$ (D). Results were compared by t-test (C) or ANCOVA test (D). *, p<0.05.
DOI: https://doi.org/10.7554/eLife.43387.016

The following source data is available for figure 6:

**Source data 1.** $CO_2/H^+$-evoked activity in chemosensitive RTN neurons in slices from $Slc32a1^{cre/+}$ and $Scn1a^{\Delta E26}$ mice.
DOI: https://doi.org/10.7554/eLife.43387.017

where respiratory control centers are located, resulted in premature death (*Cheah et al., 2012*). These results suggest respiratory dysfunction in DS is a secondary consequence of cortical seizure activity propagating to and disrupting brainstem function.

There are numerous direct and indirect projections from the cortex to brainstem respiratory centers (*Shea, 1996*) that may serve as the anatomical substrate for seizure-induced respiratory dysfunction. For example, recent work in humans showed that apnea and arterial oxygen desaturation occurred when cortical seizure activity spread to the amygdala (*Dlouhy et al., 2015*) and presumably activated descending inhibitory projections to brainstem respiratory centers. However, SUDEP can also occur in epilepsy patients in the absence of an overt seizure or outside the peri-ictal period (*Lhatoo and Shorvon, 1998*) and some epilepsy patients exhibit breathing problems including a suppressed ventilatory response to $CO_2$ (*Sainju et al., 2019*), thus suggesting factors other than acute seizures disrupt respiratory control and predispose individuals to SUDEP. For example, it is possible that repeated bombardment of brainstem respiratory centers by frequent cortical seizure events alters cellular or neural network function, leading to progressive respiratory disruption and increased SUDEP propensity. Consistent with this possibility, patients with temporal lobe epilepsy (a

common type of focal epilepsy) show widespread alterations in neural network activity including at the level of the brainstem (*Englot et al., 2018*). However, it remains unclear whether elements of respiratory control are compromised by repeated seizure activity in a similar manner.

Our results show that *Scn1a* transcript is highly expressed by brainstem inhibitory neurons and to a lesser extent by glutamatergic neurons (*Figure 1A–C*); therefore, loss of *Scn1a* function conceivably will also directly impact brainstem inhibitory neurons independent of descending seizure activity. Consistent with this possibility and analogous to cortical inhibitory neurons in $Scn1a^{-/+}$ knockout models (*Cheah et al., 2012*) and induced pluripotent stem cells derived from DS patients with an *Scn1a* truncation mutation (*Higurashi et al., 2013*), we found that inhibitory neurons in the RTN region expressed the *Scn1a* A1783V pathological variant produced fewer action potentials in response to sustained depolarizing current injection and were more prone to depolarization block compared to inhibitory neurons from control mice (*Figure 4A–D*). These results suggest that loss of Scn1a might suppress inhibitory tone in brainstem respiratory centers including the RTN where inhibitory neurons appear to interact with and regulate the activity of excitatory chemosensitive neurons (*Ott et al., 2011*).

We confirmed this possibility at the cellular level by showing that baseline activity and $CO_2/H^+$-dependent output of RTN chemoreceptors in slices from $Scn1a^{\Delta E26}$ mice was enhanced compared to RTN chemoreceptors in slices from $Slc32a1^{cre/+}$ control mice, thus demonstrating that RTN chemoreceptor function is potentiated in this DS model. Although increasing RTN chemoreceptor drive is expected to increase baseline breathing and the ventilatory response to $CO_2$, this response is dependent in inhibitory neurotransmission. For example, the frequency response elicited by photoactivation of RTN chemoreceptors in vitro was eliminated by systemically blocking inhibition with picrotoxin and strychnine (*Cregg et al., 2017*). Therefore, although RTN chemoreceptor function is perturbed in $Scn1a^{\Delta E26}$ mice, it is likely that other respiratory elements also contribute to the observed hypoventilation phenotype. For example, evidence suggests serotonergic neurons in the dorsal and medullary raphe, which modulate breathing in response to changes in $CO_2$ and arousal (*Richerson, 2004*; *Buchanan and Richerson, 2010*) are inhibited during and after seizures (*Zhan et al., 2016*). Furthermore, loss of serotonergic signaling has been shown to increase the likelihood of seizure-induced respiratory arrest in a mouse model of epilepsy, whereas administration of serotonin reuptake inhibitors does the opposite (*Tupal and Faingold, 2019*). Therefore, disruption of serotonergic signaling may contribute to breathing problems in $Scn1a^{\Delta E26}$ mice. Another possibility worth noting is that loss of *Scn1a* function may compromise inspiratory rhythm generation by the pre-bötzinger complex. This is notable because loss of inhibitory tone within this region has been shown to decrease respiratory frequency (*Del Negro et al., 2018*; *Baertsch et al., 2018*) and the breathing phenotype in $Scn1a^{\Delta E26}$ mice under high $CO_2$ conditions preferentially involves diminished respiratory frequency but otherwise normal tidal volume (*Figure 3F–H*). There are certainly several other possible mechanisms by which loss of *Scn1a* may contribute to breathing problems in DS. Results presented here represent a first step towards understanding the cellular basis of disordered breathing in this disease.

Despite the prevalence of *Scn1a* missense mutations in DS (*Parihar and Ganesh, 2013*), few studies have characterized the pathophysiology associated with specific mutant alleles. This is particularly important for the development of patient-directed therapies because the aberrant products of *Scn1a* missense mutations are potentially expressed, thus representing a novel therapeutic target that is absent from haploinsufficient models of DS. Here, we show that expression of the *Scn1a* pathological variant A1783V in inhibitory neurons results in seizures and premature death on an accelerated time scale compared to haploinsufficient DS models (*Catterall, 2012*). Inhibitory neurons from $Scn1a^{\Delta E26}$ mice showed a modest reduction in channel transcript (*Figure 1C*) that may contribute to loss of inhibitory tone (*Figures 4–5*); however, transcript containing the A1783V variant was expressed in tissue from $Scn1a^{\Delta E26}$ mice (*Figure 1A*) and the repetitive firing characteristics of inhibitory neurons from $Scn1a^{\Delta E26}$ mice is consistent with loss of function due to increased Nav1.1 channel inactivation. For example, genotype differences in the action potential amplitude and rate of depolarization were diminished under experimental conditions designed to remove $Na^+$ channel inactivation. Therefore, an effective treatment for *Scn1a* A1783V-associated pathology might be to selectively potentiate Nav1.1 channel activity by slowing voltage-dependent inactivation. However, future experiments are required to test this possibility.

In sum, our results show that expression of *Scn1a* A1783V in inhibitory neurons results in clinical features of DS including spontaneous seizures and respiratory dysfunction. At the cellular level, brainstem inhibitory neurons in the RTN of slices from $Scn1a^{\Delta E26}$ are less excitable whereas glutamatergic chemosensitive neurons are more excitable. Thus, our findings indicate that RTN chemoreceptors are a potential substrate for respiratory dysfunction in DS.

# Materials and methods

**Key resources table**

| Reagent type | Designation | Source or reference | Identifiers | Adtl. info |
|---|---|---|---|---|
| Strain, strain background (M. musculus, *Scn1a A1783V*, C57BL6/J background) | B6(Cg)-Scn1 atm1.1Dsf/J | Jackson Laboratories | RRID:IMSR_JAX:026133 | unpublished model |
| Strain, strain background (M. musculus, Vgat-iris-Cre, mixed 129/SvJ and C57BL6/J background) | Slc32a1tm2(cre)Lowl/J | PMID: 21745644 | RRID:IMSR_JAX:016962 | |
| Strain, strain background (M. musculus, tdTomato reporter Ai14, C57BL6/J background) | B6.Cg-Gt(ROSA) 26Sortm14 (CAG-tdTomato)Hze/J | PMID: 20023653 | RRID:IMSR_JAX:007914 | |
| Genetic reagent (M-MLV Reverse Transcriptase (200 U/μL)) | MMLV RT first-strand reagent | ThermoFisher Scientific | 28025013 | |
| Genetic reagent (GoTaq Flexi DNA Polymerase) | Taq polymerase | Promega | M8291 | |
| Antibody | goat anti-Phox2b antibody | R and D Systems | AF4940; RRID:AB_10889846 | used on fixed tissue, 1:500 dilution |
| Antibody | rabbit anti-Lucifer Yellow antibody | Invitrogen | A-5750; RRID:AB_2536190 | used on fixed tissue, 1:2000 dilution |
| Sequence based reagent | RNAscope Probe- Mm-Scn1a | ACDBio | 434181 | 1:50 |
| Sequence based reagent | RNAscope Probe- Mm-Slc32a1-C2 | ACDBio | 319191-C2 | 1:50 |
| Sequence based reagent | RNAscope Probe- Mm-Slc17a6-C2 | ACDBio | 319171-C2 | 1:50 |
| Commercial assay or kit | RNAscope Fresh Frozen Multiplex Fluorescent Kit | ACDBio | 320851 | |
| Commercial assay or kit | NEB PCR Cloning Kit | New England BioLabs | E1202S | |
| Commercial assay or kit | QIAprep Gel Extraction Kit | Qiagen | 28704 | |
| Commercial assay or kit | QIAprep Spin Miniprep Kit | Qiagen | 27104 | |
| Commercial assay or kit | Direct-zol RNA MicroPrep | Zymo Research | R2061 | |
| Chemical compound, drug | Lucifer Yellow | Sigma | B4261 | 0.10% |
| Software, algorithm | SnapGene Viewer | SnapGene | RRID:SCR_015053 | |

*Continued on next page*

*Continued*

| Reagent type | Designation | Source or reference | Identifiers | Adtl. info |
|---|---|---|---|---|
| Software, algorithm | Ponemah | DSI | RRID:SCR_017107 | Version 5.20 |
| Software, algorithm | Spike | Cambridge Electronic Design | RRID:SCR_000903 | Version 5.0 |
| Software, algorithm | Sirenia | Pinnacile Technology | RRID:SCR_016183 | |
| Software, algorithm | Matlab | Mathworks | RRID:SCR_001622 | Version R2018 |
| Software, algorithm | Brainstorm | *Tadel et al., 2011* | RRID:SCR_001761 | Version 3.0 |
| Software, algorithm | Prism 7 | GraphPad | RRID:SCR_002798 | Version 7.03 |
| Software, algorithm | pCLAMP 10 | Molecular Devices | RRID:SCR_011323 | Version 10 |
| Software, algorithm | ImageJ | NIH | RRID:SCR_003070 | Version 2.0.0 |

## Ethics statement

All experiments were performed according to the guidelines described in the National Institutes of Health Guide for the Care and Use of Laboratory Animals and were approved by the Institutional Animal Care and Use Committee of the University of Connecticut, Storrs (Protocols A16-034 and A17-002).

## Animals

$Scn1a^{\Delta E26}$ mice were generated by crossing offspring of $Slc32a1^{cre+/+}$ (RRID:IMSR_JAX:016962) and homozygous Gt(ROSA)26Sor$^{tm14(CAG-tdTomato)Hze}$/J reporter mice (Ai14; RRID:IMSR_JAX:007914) with heterozygous $Scn1a^{A1783Vfl/+}$ mice (RRID:IMSR_JAX:026133) to introduce the $Scn1a$ variant A1783V conditionally in inhibitory neurons. Experimental animals heterologously express both the reporter and the $Scn1a$ A1783V pathological variant ($Scn1a^{\Delta E26}$ mice) and litter mate controls and litter mate controls used for experiments were Vga$^{cre-/-}$::$Tdt^{+/-}$:: $Scn1a^{A1783Vfl/+}$ and $Slc32a1^{cre+/-}$:: $Tdt^{+/-}$::$Scn1a^{+/+}$ (on a common background of 90% *C57BL/6J and 10% 129/SvJ*). The proportion of each background stain was determined by Genome Scan Analysis performed by the Jackson Laboratory. Aged matched mice of each genotype and sex were used for all experiments included in this study.

## PCR and sequencing

Somatosensory cortex and brainstem tissue was isolated from pups of each genotype ($Scn1a^{\Delta E26}$, $Scn1a^{fl/+}$, and $Slc32a1^{cre/+}$) and triturated to make a single cell suspension for RNA isolation using the Zymo RNA Microprep kit (Zymo, Cat # R2061). The RNA was converted to cDNA using the MMLV RT first-strand reagent (ThermoFisher, Cat. # 28025013). The cDNA was then amplified using $Scn1a$ gene primers (both 5' to 3', exon 25 forward: GCATTATGTGACAAGCATTTTGTCACGC, exon 26 reverse: GCGCTCTAGAACCCCCTCTCATTTGCCAC) in a 24.5 uL reaction volume per sample containing: 5 uL of 5X buffer with loading dye (Promega M891A), 4 uL of MgCl$_2$ (25 mM), 1 uL of dNTPs (10 mM), 12.3 uL of DEPC H$_2$O, 0.2 uL of Taq polymerase (Promega M8291), 1 uL of each primer (30 pM/uL) and 0.5 uL of cDNA. The cycling protocol was 95℃ for 2 min, 95℃ for 30 s/58℃ for 1 min/72℃ for 1 min (repeated for 30 cycles total), 72℃ for 5 min, 12℃ hold. The PCR product was run on a 1.5% EtBr gel at 90 mV for 30 min.

The amplified 831 bp product was excised and placed into a new DNase/RNase free micro-centrifuge tube. To extract the amplified cDNA we used the Qiagen Gel Extraction Kit (Cat 28704). The purified sequence was inserted into the linearized pMiniT 2.0 vector using the NEB PCR cloning kit and NEB 10-beta competent cells (Cat E1202S). The resulting transformed competent cells were then streaked out onto LB agar plates with ampicillin and grown out overnight at 37℃. Single colonies were selected and placed into 2 mL of LB broth with ampicillin in a 5 mL polystyrene round

bottom tube and grown out for 16 hr/overnight, shaken at 250 rpm at 37°C. 1.5 mL of each bacterial sample was used for plasmid purification using the Qiagen Spin Miniprep Kit (Cat 27104) according to manufacture instructions. Samples were then sequenced at Eurofins Genomics using the forward or reverse primers for the pMiniT 2.0 vector provided in the NEB PCR cloning kit. Once sequences were returned from Eurofins Genomics, the SnapGene Viewer (RRID:SCR_015053) was used to map the sequence to the pMini.T 2.0 vector map; the sequence was then searched in NCBI BLAST to yield similar sequences from the mouse genome. All samples mapped back to the *Scn1a* gene, between exon 25 and 26. After mapping the inserted sequence to *Scn1a*, the A to V single nucleotide mutation was localized and identified at nucleotide 1772 mapped to the mouse genome. In total, a minimum of 15 samples were collected from each genotype for sequencing.

## Fluorescent in situ hybridization (FISH)

To prepare fresh frozen slice, postnatal week 2 mice of both genotypes were anesthetized with isoflurane, decapitated, and brainstem tissues were rapidly frozen with dry ice and embedded with OCT compound. Brainstem slices (14 um thick) containing the RTN were crysectioned and collected onto SuperFrost Plus microscope slides. Slices were fixed with 4% paraformaldehyde and dehydrated with 50%,70% and 100% ethanol. FISH was processed with the instruction of RNAscope Multiplex Fluorescent Assay (ACD, 320850), the probes used in our study were designed and validated by ACD (*Table 3*). Confocal images of FISH experiments were obtained using a Leica TSC Sp8 and confocal image files containing image stacks were loaded into ImageJ (version 2.0.0, NIH, RRID: SCR_003070).

## Unrestrained whole-body plethysmography

Respiratory activity was measured using a whole-body plethysmograph system (Data Scientific International; DSI), utilizing animal chamber (600 mL volume) maintained at room temperature and ventilated with air (1.3 L/min) using a small animal bias flow generator. Fifteen day old mice (~7 g) were individually placed into a chamber and allowed 2 hr to acclimate prior to the start of an experiment. Respiratory activity was recorded using Ponemah 5.20 software (DSI) for a period of 15 min in room air followed by exposure to graded increases in $CO_2$ from 0% to 7% $CO_2$ (balance $O_2$). Body temperature was measured before and after each experiment and although body temperature tended to drop ~1 °C by the end of an experiment, there were no genotype difference in the degree of cooling (p=0.37). Parameters of interests include respiratory frequency ($F_R$, breaths per minute), tidal volume ($V_T$, measured in mL; normalized to body weight and corrected to account for chamber and animal temperature, humidity, and atmospheric pressure), and minute ventilation ($V_E$, mL/min/g). A 20 s period of relative quiescence after 2 min of exposure to each condition was selected for analysis. Spontaneous apneic events, conservatively defined as three or more missed breaths not preceded by a sigh or augmented breath, were analyzed off-line. All experiments were performed between 9 a.m. and 6 p.m. to minimize potential circadian effects.

## Acute slice preparation and in vitro electrophysiology

Slices containing the RTN were prepared as previously described (*Mulkey et al., 2007*). In short, rats were anesthetized by administration of ketamine (375 mg/kg, I.P.) and xylazine (25 mg/kg; I.P.) and rapidly decapitated; brainstems were removed and transverse brain stem slices (300 μm) were cut using a microslicer (DSK 1500E; Dosaka) in ice-cold substituted Ringer solution containing the following (in mM): 260 sucrose, 3 KCl, 5 $MgCl_2$, 1 $CaCl_2$, 1.25 $NaH_2PO_4$, 26 $NaHCO_3$, 10 glucose, and one kynurenic acid. Slices were incubated for 30 min at 37°C and subsequently at room temperature in a normal Ringer's solution containing (in mM): 130 NaCl, 3 KCl, 2 $MgCl_2$, 2 $CaCl_2$, 1.25

**Table 3.** Probes used for FISH.

| Gene name | Probe cat no. | Target region |
|---|---|---|
| *Scn1a* | 434181 | 1624–2967 |
| *Slc32a1* | 319191-C2 | 894–2037 |
| *Slc17a6* | 319171-C2 | 1986–2998 |

DOI: https://doi.org/10.7554/eLife.43387.018

NaH$_2$PO$_4$, 26 NaHCO$_3$, and 10 glucose. Both substituted and normal Ringer's solutions were bubbled with 95% O$_2$ and 5% CO$_2$ (pH = 7.30).

Individual slices containing the RTN were transferred to a recording chamber mounted on a fixed-stage microscope (Olympus BX5.1WI) and perfused continuously (~2 ml/min) with a bath solution containing (in mM): 140 NaCl, 3 KCl, 2 MgCl$_2$, 2 CaCl$_2$, 10 HEPES, 10 glucose (equilibrated with 5% CO$_2$; pH = 7.3). All recordings were made with an Axopatch 200B patch-clamp amplifier, digitized with a Digidata 1322A A/D converter, and recorded using pCLAMP 10.0 software (RRID:SCR_011323). Recordings were obtained at room temperature (~22° C) with patch electrodes pulled from borosilicate glass capillaries (Harvard Apparatus, Molliston, MA) on a two-stage puller (P-97; Sutter Instrument, Novato, CA) to a DC resistance of 5–7 MΩ when filled with a pipette solution containing the following (in mM): 125 K-gluconate, 10 HEPES, 4 Mg-ATP, 3 Na-GTP, 1 EGTA, 10 Na-phosphocreatine (uM), 0.2% Lucifer yellow (pH 7.30). Electrode tips were coated with Sylgard 184 (Dow Corning, Midland, MI).

The firing response of chemosensitive RTN neurons to CO$_2$(3–10% CO$_2$) was assessed in the cell-attached voltage-clamp configuration (seal resistance >1 GΩ) with holding potential matched to the resting membrane potential ($V_{hold}$ = −60 mV) and with no current generated by the amplifier ($I_{amp}$ = 0 pA). Firing rate histograms were generated by integrating action potential discharge in 10 to 20 s bins using Spike 5.0 software (RRID:SCR_000903). We confirmed that all chemosensitive RTN neurons included in this study were immunoreactive for the transcription factor Phox2b.

To characterize action potential properties and repetitive firing behavior of inhibitory neurons, we made whole-cell current-clamp recordings from fluorescently labeled neurons located in the region of the RTN in slices from Vgat::TdTomato mice. Repetitive firing responses to 1 s depolarizing current steps from 0 to 300 pA (Δ 20 pA increments) were characterized from an initial holding potential of −80 mV. Action potential amplitude, threshold (dV/dT > 10 mV/mS) and the maximum rate of depolarization obtained from the peak of the first time derivative of the action potential were characterized for spontaneous spikes measured under resting conditions (holding current = 0 pA) and for the first spike elicited after delivering a positive (+200 pA) or negative (−100 pA) 1 s current step. All whole-cell recordings had an access resistance (Ra) <20 MΩ, recordings were discarded if Ra varied >10% during an experiment. A liquid junction potential of −14 mV was accounted for during each experiment.

## Immunofluorescence staining

Slices were fixed in 4% PFA over night after recording, and blocked with 5% normal horse serum in 1X PBS with 2.5% triton for 1 hr. Slices were incubated in goat anti-phox2b antibody (RRID:AB_10889846) and rabbit anti-lucifer yellow antibodies (RRID:AB_2536190) mixed in blocking solution under 4°C overnight. After washing the primary antibody a secondary antibody was applied for 2 hr followed by an additional wash and mounting with ProLong Gold Antifade Reagent (Invitrogen, P36934). Slices were imaged using a Leica SP8 confocal microscope (40x/1.3 HC oil objective) to identify cells that co-express Alex Fluor 647 for phox2b and Alex Fluor 488 for lucifer yellow.

## Electrocorticography recording

Subdural EcoG electrodes were implanted in 15 day old *Slc32a1*$^{cre/+}$ and *Scn1a*$^{ΔE26}$ mouse pups. To minimize damage we used stainless steel wire electrodes (diameter = 0.003 in) (A-M system, 790900) inserted just under the skull for a length of 2 mm into each hemisphere near the fontal cortex. A reference wire electrode was placed in the posterior cortex. Each electrode was connected to a Mill-MAX miniature socket (digikey, ED11265-ND) and secured to the skull with super glue. Differential voltage signals were amplified 1000 × with a DAM-50 differential amplifier (1 Hz low filter, 10 kHz high filter), digitized at 5 kHz and recorded using Sirenia Software (RRID:SCR_016183).

Mice were allowed to 12 hr to recover from surgery before recording EcoG activity for a period of 2 hr. We also video recorded all experiments to correlate animal behavior with EcoG recordings. Only spike wave discharge (SWD) activity that occurred in conjunction with observable seizure behavior was included in the analysis. Any data including movement artifacts were excluded from analysis. The same criteria for seizure events were used for both mutant mice and control group. The full duration of each seizure event was segmented and then down sampled from 600 Hz to 100 Hz to focus on the frequency range of interest (0–50 Hz) prior to performing the power spectral analysis

in Matlab (RRID:SCR_001622). Frequency ranges of EcoG signals are defined as follows: delta, δ (1–5 Hz), theta, Θ (6–8 Hz), alpha, α (9–16 Hz), beta, β (17–36 Hz), and gamma, γ (37–50 Hz). Frequency analysis results were normalized to the maximum frequency amplitude at each event. For each frequency range, maximal amplitude and area under each frequency range were calculated to report the spectral power. To show the time-varying frequency distribution, time frequency analysis using Hilbert and Morlet transformations were also performed using Brainstorm 3.0 (*Tadel et al., 2011*, RRID:SCR_001761).

### Seizure behavior scoring

We video monitored mice for 1 hr after placing them individually in a cage and giving them access to food and water ad lib. Seizure behavior during this time was evaluated using the Racine scoring system as follows: score 1, mouth and facial movements; score 2, head nodding; score 3, forelimb clonus; score 4, rearing with limb clonus; score 5, full body clonus, rearing and falling.

### Thermal seizure induction

To record febrile seizures mice were placed in a Plexiglas cylindrical chamber and we continuously monitor animal body temperature using a Type T thermocouple rectal probe connected to a feedback temperature controller and a heating lamp (Physitemp) that was positioned directly above the chamber. This system allowed us to maintain body temperature to within ±0.4°C of command temperature. Mice were held at 37°C for 10 min before body temperature was increased in 0.5°C increments every 2 min until a tonic-clonic seizure occurs or 42°C is reached. All experiments were video recorded for later conformation of seizure behavior.

### Statistical analysis

Data are reported as mean ± SE. All experiments were performed blind to genotype and all statistical analysis was performed using Prism 7 (RRID:SCR_002798). Power analysis was used to determine sample size, all data sets were tested for normality using Shapiro-Wilk test, and comparisons were made using t-test, Chi Square test, Fisher's exact test, one-way or two-way ANOVA followed by multiple comparison tests as appropriate. The specific test used for each comparison is reported in the figure legend and all relevant values used for statistical analysis are included in the results section.

## Acknowledgements

We thank Drs. Ana Mingorance (Chief Development Officer of the Loulou Foundation), Rahul Kanadia and Anastasios Tzingounis (Dept. Physiology and Neurobiology, Univ. Connecticut) for their constructive suggestions regarding this project. This work was supported by funds from the National Institutes of Health Grants HL104101 (DKM), HL137094 (DKM) NS104999 (JLL) and F31HL142227 (CMC). Additional funds were also provided by the Dravet syndrome Foundation Grant AG180243 (DKM) and American Epilepsy Society (F-SK).

## Additional information

### Funding

| Funder | Grant reference number | Author |
| --- | --- | --- |
| National Institutes of Health | HL104101 | Daniel K Mulkey |
| Dravet Syndrome Foundation | AG180243 | Daniel K Mulkey |
| American Epilepsy Society | | Fu-Shan Kuo |
| National Institutes of Health | NS104999 | Joseph J LoTurco |
| National Institutes of Health | F31HL142227 | Colin M Cleary |
| National Institutes of Health | HL137094 | Daniel K Mulkey |

The funders had no role in study design, data collection and interpretation, or the decision to submit the work for publication.

## Author contributions
Fu-Shan Kuo, Conceptualization, Data curation, Formal analysis, Writing—original draft, Project administration, Writing—review and editing; Colin M Cleary, Data curation, Formal analysis, Funding acquisition, Writing—review and editing; Joseph J LoTurco, Supervision, Methodology, Writing—review and editing; Xinnian Chen, Formal analysis, Methodology, Writing—review and editing; Daniel K Mulkey, Conceptualization, Supervision, Funding acquisition, Writing—original draft, Project administration, Writing—review and editing

## Author ORCIDs
Colin M Cleary http://orcid.org/0000-0003-0305-1324
Daniel K Mulkey http://orcid.org/0000-0002-7040-3927

## Ethics
Animal experimentation: All animal use was in accordance with guidelines approved by the University of Connecticut Institutional Animal Care and Use Committee. (Protocols A16-034 and A17-002).

## Decision letter and Author response
Decision letter https://doi.org/10.7554/eLife.43387.021
Author response https://doi.org/10.7554/eLife.43387.022

## Additional files

### Supplementary files
• Transparent reporting form
DOI: https://doi.org/10.7554/eLife.43387.019

### Data availability
We have included source data files for all summary figures that do not include individual data points.

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
