## [Decision Letter]

Thank you for submitting your article "Disordered breathing in a mouse model of Dravet syndrome" for consideration by *eLife*. Your article has been reviewed by two peer reviewers, and the evaluation has been overseen by Jan-Marino Ramirez as the Reviewing Editor, and Ronald Calabrese as the Senior Editor. The reviewers have opted to remain anonymous.

The reviewers have discussed the reviews with one another and the Reviewing Editor has drafted this decision to help you prepare a revised submission.

Summary:

Dravet syndrome (DS) is a severe neurodevelopmental disorder largely due to heterozygous mutation of SCN1A encoding Nav1.1; patients exhibit temperature-sensitive seizures with onset in infancy, ataxia, developmental delay, features of autism spectrum disorder, and SUDEP, and *Scn1a*^+/-^ mice recapitulate the core features of the disorder. In fact, DS may have the highest SUDEP rate among the epilepsies. However, the mechanisms of SUDEP remain unknown.

Kuo and colleagues used a recently developed transgenic mouse carrying a floxed stop *Scn1a* A1783V allele to generate a mouse model of DS with the disease-causing *Scn1a* A1783V mutation restricted to inhibitory neurons. Using electrophysiology and plethysmography, the authors show disordered breathing and abnormal function of neurons in the respiratory brainstem in this model. The cellular neurophysiology in a challenging preparation is of the highest quality and it is complemented by a careful characterization of the respiratory physiology. The authors show that mice expressing *Scn1a*-p.A1783A in interneurons show decreased respiratory rate, lower tidal volume, and lower minute ventilation, with an impaired capacity to increase respiration in response to C02. Inhibitory interneurons in the RTN, which are shown to express Nav1.1, reveal lower spontaneous firing rates and decreased firing frequency in response to depolarizing current injections despite identical passive membrane properties. The authors also demonstrate an enhanced activity of glutamatergic chemosensory neurons in the RTN, presumably as a result of the disinhibition. These findings demonstrate that the SCN1A A1783V mutation, which is linked to DS in humans, causes epilepsy, SUDEP, and respiratory dysfunctions in mice, even when restricted to inhibitory interneurons alone. The reviewers raised several concerns that require further clarification as specified in the essential revisions as detailed below.

Essential revisions:

1) One major issue with this manuscript is the studied mouse and its genetic background. First, this *Scn1a*-p.A1783A line has not previously been published, at least so far as this reviewer is aware. The authors state repeatedly that the phenotype of this line is similar to other lines, but the mortality rate appears to be much more severe, and the mechanism of death is unclear and seems likely to be different. Added burden is placed on the authors to characterize this line, and rigor is required with regard to maintenance of consistent mouse genetic background. Do the mice have temperature-sensitive seizures? The EEG data is confusing with regard to what the authors are calling seizures vs. interictal discharges. The authors need to apply a widely accepted definition of a mouse EEG seizure and just tell us what the seizure frequency is. Why are the wild-type control mice having seizures?

2) It is well known that mouse behavioral phenotypes, including epilepsy, and in particular, the phenotype of *Scn1a*^+/-^ mice, is dependent upon mouse genetic background. Here, the authors use conditional *Scn1a*-p.A1783A mice crossed to a VGAT-Cre driver line. These VGAT-Cre mice are on a mixed C57:129S6 background. For a subset of experiments, it appears that the authors cross this mouse to a tdTomato reporter strain, which is on a C57 background. Hence, the *Scn1a*-p.A1783A/VGAT-Cre mice are on a different background than the *Scn1a*-p.A1783A/VGAT-Cre.tdT mice. This needs to be clearly stated and may require additional controls. Already, the abstract indicates that mice used in this study had a mixed background, but then this issue is not elaborated upon in the text.

3) Figure 1F suggests that the A1783V allele does not express, only the WT allele does. If so, it is unclear how the A1783V variant will contribute to the loss-of-function via impairment of channel inactivation as described in Figure 5, considering that the protein is absent. Is the *Scn1a* A1783V partially expressed? A discussion of this issue will be important. The electrophysiological changes observed could be associated with either or both reduced expression or impaired inactivation of the channel.

4) It is unclear why control mice exhibit behavioral and EEG seizures (Figure 2). Is the A1783V mutation activated in the absence of Cre recombinase in some cases?

5) The reviewers do not agree with the authors' assertion that "this mouse model presents with a respiratory phenotype strikingly similar to that exhibited by DS patients." A peri-ictal respiratory phenotype has been shown in some DS patients. Here, the authors show a dramatic baseline respiratory phenotype and 100% early death that may be from non-seizure related causes. Thus the authors need to clarify these differences and should alter their statement, accordingly.

---

## [Author Response]

Essential revisions:1) One major issue with this manuscript is the studied mouse and its genetic background. First, this Scn1a-p.A1783A line has not previously been published, at least so far as this reviewer is aware. The authors state repeatedly that the phenotype of this line is similar to other lines, but the mortality rate appears to be much more severe, and the mechanism of death is unclear and seems likely to be different. Added burden is placed on the authors to characterize this line, and rigor is required with regard to maintenance of consistent mouse genetic background.

We appreciate that we need to do a more thorough job describing the background genetics and phenotype of this new model. In particular, we crossed homozygous Gt(ROSA)26Sor^tm14(CAG-tdTomato)Hze^/Jreportermice (Ai14; JAX no. 007914) on a C57BL/6J background with homozygous Vgat^Cre^ mice (JAX no. 016962) on a mixed background of 75% C57BL6/J: : 25% 129/SvJ to produce Vgat-Cre.tdT double-heterozygous mice with a 85% C57BL6/J: : 15% 129/SvJ background. These mice were crossed with Scn1a^A1783Vfl/+^ (Jax no. 026133) maintained on a pure C57BL/6J background to produce experimental animals with a genotype of *Vgat^cre+/-^::TdT^+/-^::Scn1a^fl/+^*(*Scn1a^ΔE26^*) and control animals of the following genotypes *Vgat^cre-/-^::TdT^+/-^::Scn1a^fl/+^*(*Scn1a^fl/+^*) and *Vgat^cre+/-^::TdT^+/-^::Scn1a^+/+^ (Vgat^cre/+^*). Experimental and control mice had a common background of 90% C57BL6/J: : 10% 129/SvJ. The proportion of each background stain was determined by Genome Scan Analysis performed by the Jackson Laboratory. These details have been added to the text and we included a breeding scheme as Figure 1—figure supplement 1.

Do the mice have temperature-sensitive seizures?

Great suggestion! We have characterized febrile seizure propensity in Scn1a^ΔE26^ mice and Vgat^cre+/-^ litter mate controls (mixed sex P12-14). We found that all Scn1a^ΔE26^ mice (n=9) develop tonic-colonic seizures at an average body temperature of 41.09 ± 0.19 ◦C. Conversely, none of the Vgat^cre+/-^ litter mate controls (n=10) show seizure activity up to the cut-off temperature of 42.5◦C. These new results have been included as a new Table 2 and discussed in the text. We also added these experimental details to the Materials and methods. Note that unfortunately our colonies have had very few litters recently and so we are unable to report febrile seizure activity in *Scn1a^fl/+^*control animals.

The EEG data is confusing with regard to what the authors are calling seizures vs. interictal discharges. The authors need to apply a widely accepted definition of a mouse EEG seizure and just tell us what the seizure frequency is. Why are the wild-type control mice having seizures?

Sorry for this confusion. In the original version of this manuscript, we included any large amplitude spike activity in our analysis and in doing so unintentionally included behavioral artifacts. We have now redefined seizure-like ECoG activity as poly-spike bursts rather than spike wave discharges because the latter term is typically used to refer to absence seizure activity (PMID: 24861780). We have reanalyzed our data found that poly-spike activity that lasted a minimum of 14 ms consistently occurred in conjunction with obvious seizure activity (Racine score > 3). Based on this, we define epileptic spike activity for this model as abrupt onset poly-spiking events with greater than twice baseline amplitude, minimum duration of 14 ms, and that occur in conjunction with seizure activity (category 3-5). Based on this criteria, the majority of *Vgat^cre/+^*control animals did not show any seizure like activity. However, we did observe three Vgat^cre/+^ control mice that each showed one epileptic event. The reason for this is unclear but may involve potential confounding effects caused by surgical placement of the electrodes or unanticipated background strain issues. In any case, the background stain was consistent for all experimental and control animals.

We have modified the Results section to make these points more clear. We also added a new panel to Figure 2 (panel Bi) to illustrate the duration distribution of all poly-spike events of each mouse included in this analysis. This new analysis strengthened our conclusion that *Scn1a^ΔE26^*mice have a severe seizure phenotype.

2) It is well known that mouse behavioral phenotypes, including epilepsy, and in particular, the phenotype of Scn1a+/- mice, is dependent upon mouse genetic background. Here, the authors use conditional Scn1a-p.A1783A mice crossed to a VGAT-Cre driver line. These VGAT-Cre mice are on a mixed C57:129S6 background. For a subset of experiments, it appears that the authors cross this mouse to a tdTomato reporter strain, which is on a C57 background. Hence, the Scn1a-p.A1783A/VGAT-Cre mice are on a different background than the Scn1a-p.A1783A/VGAT-Cre.tdT mice. This needs to be clearly stated and may require additional controls. Already, the abstract indicates that mice used in this study had a mixed background, but then this issue is not elaborated upon in the text.

As noted in our response to the first concern, we have provided additional details regarding the background genetics of each strain used in these experiments. The proportion of each stain was determined by Genome Scan Analysis performed by the Jackson Laboratory. We have also added a supplementary figure to illustrate the breeding scheme (Figure 1—figure supplement 1).

3) Figure 1F suggests that the A1783V allele does not express, only the WT allele does. If so, it is unclear how the A1783V variant will contribute to the loss-of-function via impairment of channel inactivation as described in Figure 5, considering that the protein is absent. Is the Scn1a A1783V partially expressed? A discussion of this issue will be important. The electrophysiological changes observed could be associated with either or both reduced expression or impaired inactivation of the channel.

Thanks for this suggestion. New sequencing results show that *Scn1a* transcript containing the A1783V mutation was detectable in 8 of 20 (40%) sequences tested, thus confirming at the mRNA level that the mutant channel is expressed in experimental animals (*Scn1a^Δ26^*). These new results have been added to Figure 1. However, our fluorescent in situ hybridization results shown in Figures 1B-C suggest inhibitory neurons from *Scn1a^ΔE26^* mice have slightly reduced levels of Scn1a transcript. Therefore, as suggested by the reviewer, we have modified the text to make clear that diminished channel expression or expression of channel with altered properties may contribute to the electrophysiological changes observed in slices from Scn1a^ΔE26^ animals.

4) It is unclear why control mice exhibit behavioral and EEG seizures (Figure 2).

See our second response to concern 1 above. In short, we provide a complete description of seizure like behavior and ECoG activity for Vgat^cre/+^ and Scn1a^ΔE26^ lines. Based on standard criteria, we found that control animals showed minimal seizure-like activity both in terms of behavior (Table 1) and poly-spike activity (Figure 2). We cannot say the Vgat^cre/+^ controlswere completely devoid of seizure like behavior and factors including trauma during electrode placement or background strain specific issues may be contributing factors.

Is the A1783V mutation activated in the absence of Cre recombinase in some cases?

Excellent point! We failed to detect any mutant transcript in 20 samples from *Scn1a^A1783Vfl/+^*tissue. As a positive control, we did detect A1783V in samples from *Scn1a^ΔE26^*. Therefore, these results suggest there is minimal leaky mutant expression in the absence of cre. These new results have been added to Figure 1.

5) The reviewers do not agree with the authors' assertion that "this mouse model presents with a respiratory phenotype strikingly similar to that exhibited by DS patients." A peri-ictal respiratory phenotype has been shown in some DS patients. Here, the authors show a dramatic baseline respiratory phenotype and 100% early death that may be from non-seizure related causes. Thus the authors need to clarify these differences and should alter their statement, accordingly.

To clarify, DS patients show baseline respiratory problems including hypoventilation, apnea and a diminished ventilatory response to CO_2_ (PMID: 29329111). This phenotype is similar to what we report for *Scn1a^ΔE26^*(Figure 3). However, as noted by the reviewer, our model develops DS like pathological features including spontaneous and febrile seizures as well as premature death at an earlier developmental time point than other DS models. We have clarified this in the text and altered our statements accordingly.